# ZeCO: Zero Communication Overhead Sequence Parallelism for Linear Attention

**Yuhong Chou**[1]*, **Zehao Liu**[1]*, **Ruijie Zhu**[3], **Xinyi Wan**[4], **Tianjian Li**[2],
**Congying Chu**[5], **Qian Liu**[2]†, **Jibin Wu**[1]†, **Zejun Ma**[2]

[1]The Hong Kong Polytechnic University
[2]TikTok
[3]UC Santa Cruz
[4]National University of Singapore
[5]Institute of Automation, Chinese Academy of Sciences

## Abstract

Linear attention mechanisms deliver significant advantages for Large Language Models (LLMs) by providing linear computational complexity, enabling efficient processing of ultra-long sequences (e.g., 1M context). However, existing Sequence Parallelism (SP) methods, essential for distributing these workloads across devices, become the primary bottleneck due to substantial communication overhead. In this paper, we introduce ZeCO (Zero Communication Overhead) sequence parallelism for linear attention models, a new SP method designed to overcome these limitations and achieve end-to-end near-linear scalability for long sequence training. For example, training a model with a 1M sequence length across 64 devices using ZeCO takes roughly the same time as training with an 16k sequence on a single device. At the heart of ZeCO lies All-Scan, a new collective communication primitive. All-Scan provides each SP rank with precisely the initial operator state it requires while maintaining a minimal communication footprint, effectively eliminating communication overhead. Theoretically, we prove the optimality of ZeCO, showing that it introduces only negligible time and space overhead. Empirically, we compare the communication costs of different sequence parallelism strategies and demonstrate that All-Scan achieves the fastest communication in SP scenarios. Specifically, on 256 devices with an 8M sequence length, ZeCO achieves a 60% speedup compared to the current state-of-the-art (SOTA) SP method. We believe ZeCO establishes a clear path toward efficiently training next-generation LLMs on previously intractable sequence lengths.

## 1 Introduction

Long-context capabilities are becoming increasingly critical for Large Language Models (LLMs), powering advancements in document-level reasoning, multimodal understanding, and retrieval-augmented generation where extensive context is paramount [1, 2, 3, 4, 5, 6]. The trajectory from models like GPT-3.5 (4K context) [7] to Gemini 1.5 Pro [8] (1M context) highlights this trend. However, pre-training models on such vast sequence lengths presents significant computational and communication challenges. Standard self-attention mechanisms exhibit quadratic complexity ($O(L^2)$) with respect to sequence length $L$. Scaling from 4k to 128k tokens, for instance, inflates the attention FLOPs by over $1000\times$. The prohibitive computational cost of training on ultra-long sequences restricts the long-context pretraining to a specialized adaptation phase (i.e., mid-training) [9], rather than allowing for extensive training with long sequences from scratch [10, 11, 12, 13, 14, 15].

---

*Equal contribution. Emails: yuhong.chou@connect.polyu.hk; zehaoliu@polyu.edu.hk.
†Corresponding authors: qian.liu@tiktok.com; jibin.wu@polyu.edu.hk.

39th Conference on Neural Information Processing Systems (NeurIPS 2025).

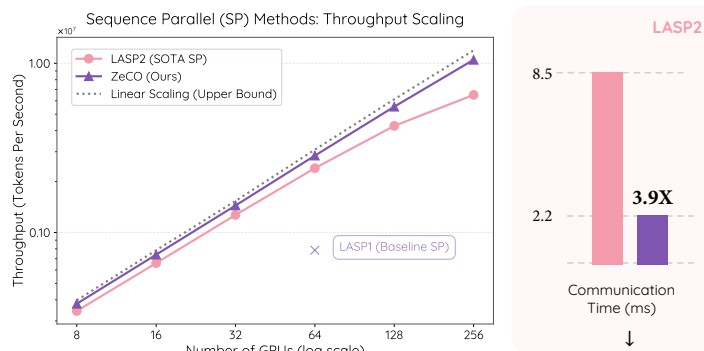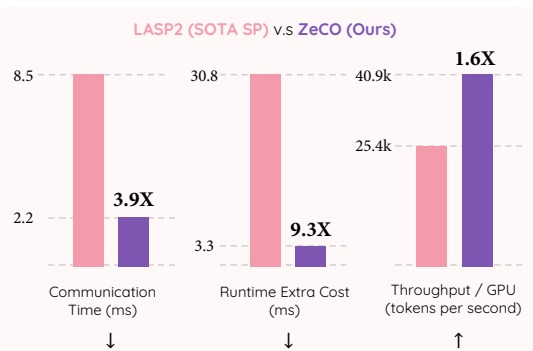

Figure 1: **(Left)** ZeCO demonstrates near-linear scaling efficiency when scaling sequence length proportionally with device count, approaching the theoretical upper bound. **(Right)** ZeCO substantially outperforms SOTA SP methods across three performance metrics: communication time, runtime extra cost, and per-GPU throughput. Metrics obtained using 256 GPUs (8M sequences) for communication / throughput and 128 GPUs (4M sequences) for runtime extra cost.

Linear attention models [16] offer an algorithmic solution by replacing the $O(L^2)$ softmax attention with operations linear in sequence length, typically $O(Ld^2)$ where $d$ is the hidden dimension. The efficiency is achieved by compressing the Key-Value (KV) cache, which would otherwise grow with sequence length, into a fixed-size hidden state representation. This method effectively eliminates the computational bottleneck of quadratic complexity and balances the per-token computation across the sequence, enabling efficient processing of ultra-long sequences. While linear attention provides these algorithmic advantages, Sequence Parallelism (SP), essential for distributing such computationally intensive workloads, paradoxically becomes a bottleneck that impedes efficient scaling across multiple devices. Existing approaches [17, 18, 19]) are often hampered by issues such as serial execution dependencies or, more significantly, substantial communication overheads that scale poorly. Consequently, despite the inherent efficiency of linear attention, the practical implementation of SP, particularly its communication burden, has become the primary impediment to achieving high throughput and true scalability for long-context training.

In this paper, we introduce ZeCO, a novel sequence parallelism strategy designed to overcome the limitations of current methods, particularly for linear attention. ZeCO achieves SOTA scalability through a fundamental redesign of its communication algorithm and communication-computation scheduling. At the heart of ZeCO lies All-Scan, a new collective operator executing a pipelined receive-scan-send pattern across devices. This new communication primitive achieves the theoretically minimal communication volume for SP. Crucially, empirical results confirm that All-Scan consistently delivers the lowest latency among all existing SP communication techniques. To further minimize communication overhead, ZeCO intelligently schedules All-Scan to overlap with local device computation, thereby enabling parallel use of both communication and computation resources. Moreover, ZeCO meticulously optimizes the auxiliary computations inherent in SP, effectively reducing their associated I/O and computational overheads to a negligible level. In summary, our main contributions can be summarized as follows.

1. We introduce ZeCO, a novel sequence parallelism method for linear attention models. ZeCO reformulates sequence parallelism by leveraging our All-Scan collective communication, which employs pipelined communication to achieve the theoretically minimum communication volume. This integrated approach enables efficient overlap of communication and computation, incurring minimal extra computational and I/O overhead.

2. We provide a comparative time-cost analysis of different sequence parallelism strategies demonstrates that ZeCO constitutes the minimum required cost, establishing its efficiency.

3. Comprehensive multi-level experiments (collective communication, operator, and model) demonstrate the significant performance gains of ZeCO. As shown in Figure 1, the All-Scan collective achieves up to $3.9\times$ communication speedup, the fastest existing sequence parallelism method, while the ZeCO sequence parallel operator delivers up to $9.3\times$ overall speedup. At the model level, ZeCO boosts throughput by over 60% and demonstrates near-linear scalability from 8 to 256 devices, even with context lengths up to 8M tokens.

## 2 Background & Related Work

In this section, we firstly provide a brief background on Gated Linear Attention (GLA), a general linear attention operator that encompasses a family of linear attention mechanisms [20, 21, 22, 23, 24, 25]. Then, we introduce the existing sequence parallelism methods.

We use bold upper-case letters (e.g., $\mathbf{Q}$) to denote matrices, And the same alphabet to represent rows of a matrix, such that $\mathbf{Q}_t$ refers to the $t$-th row of $\mathbf{Q}$. Unless otherwise specified, $p$ denotes the number of devices, $L$ denotes the sequence length per device, $d$ denotes the hidden dimension, $h$ denotes the number of attention heads, and $C$ denotes the chunk length.

### 2.1 Recurrent and Chunk-wise Form of General Linear Attention

The linear attention mechanism uses the kernel trick to remove the softmax computation in full attention and exchange the calculation order to reduce the attention computational complexity from quadratic to linear [16, 26]. There are many ways to implement this mechanism. Sevaral works [27, 22, 28] has summarized a unified form of (diagonal decay) linear model. In this paper, we use gated linear attention (GLA) operator [27], one of the generalization forms of linear models, to demonstrate our algorithm. The attention state is updated recurrently as:

$$\mathbf{S}_t = (\boldsymbol{\alpha}_t^\top 1) \odot \mathbf{S}_{t-1} + \mathbf{K}_t^\top \mathbf{V}_t, \ \mathbf{O_t} = \mathbf{Q_t}\mathbf{S_t}, \tag{1}$$

where $\boldsymbol{\alpha}_t^\top \in (0,1)^{d_k}$ is decay factor. To enable efficient parallelism during training, the sequence is partitioned into $N$ chunks of length $C$, and the recurrence is reformulated in a chunkwise manner. Let chunk $i$ include tokens from $iC$ to $(i+1)C - 1$, with decay vectors $\boldsymbol{\alpha}_{iC+j}$. Let $\mathbf{S}_{[i]} \in \mathbb{R}^{d \times d}$ be the chunk-level hidden state after processing $i$ chunks, i.e., $\mathbf{S}_{[i]} := \mathbf{S}_{iC}$. GLA define, Cumulative decay for the chunk: $\boldsymbol{\gamma}_{[i]} = \prod_{j=1}^{C} \boldsymbol{\alpha}_{iC+j}$, Token-wise scaling: $\boldsymbol{\Gamma}_{iC+j} = \frac{\boldsymbol{b}_{(i+1)C}}{\boldsymbol{b}_{iC+j}}$, $\boldsymbol{\Lambda}_{iC+j} = \frac{\boldsymbol{b}_{iC+j}}{\boldsymbol{b}_{iC}}$, where $\boldsymbol{b}_t = \prod_{s=1}^{t} \boldsymbol{\alpha}_s$. The chunk-level GLA state and output are calculated as:

$$\mathbf{S}_{[i]} = (\boldsymbol{\gamma}_{[i]}^\top 1) \odot \mathbf{S}_{[i-1]} + (\mathbf{K}_{[i]} \odot \boldsymbol{\Gamma}_{[i]})^\top \mathbf{V}_{[i]}, \tag{2}$$

The output of each chunk should be computed as:

$$\mathbf{O}_{[i]} = \underbrace{(\mathbf{Q}_{[i]} \odot \boldsymbol{\Lambda}_{[i]}) \cdot \mathbf{S}_{[i-1]}}_{\mathbf{O}_{[i]}^{\text{inter}}} + \underbrace{\left[\left((\mathbf{Q}_{[i]} \odot \boldsymbol{\Lambda}_{[i]}) \cdot (\mathbf{K}_{[i]} \odot \boldsymbol{\Gamma}_{[i]})^\top\right) \odot \mathbf{M}\right] \cdot \mathbf{V}_{[i]}}_{\mathbf{O}_{[i]}^{\text{intra}}}, \tag{3}$$

the inter-chunk recurrently updates the global state, while the intra-chunk term handles mask attention computation on the diagonal.

### 2.2 Sequence Parallelism for Linear Attention Models

**LASP1** [17] adopts a chunkwise parallelization strategy by dividing the input sequence into multiple contiguous chunks and evenly distributing them across devices. Each device serially computes the output of each device based on a linear attention formula. For communication, each device receives the state from the previous block and updates it before passing it to the next device. Although this avoids redundant communication volume, it enforces a strict serial order to be executed across devices, causing the total computation time to grow linearly with the number of devices, which severely limits parallel efficiency and throughput.

**LASP2** [18, 19] follows a similar chunkwise computation structure, but replaces the serial state passing with All-Gather communication. Each device must first collect the local states from all other devices and subsequently perform an identical scan operation on the same data. This enables devices to do computation parallelism, but introduces substantial communication overhead. The total communication volume grows linearly with the number of devices, as each device must collect state tensors from all others.

### 2.3 Sequence Parallelism for Full Attention

The sequence parallelism in Megatron-LM [29] (also known as Context Parallelism) and Ring Attention [30, 31] achieve global synchronization of KV blocks in either an all-at-once or pipelined manner, based on All Gather and P2P communication respectively, and have been widely adopted [10, 18]. Ulysses [32]distributes the computation of different self-attention heads across devices, which is easy to implement but incompatible with tensor parallelism (TP) and limited by the number of heads. Ring Self Attention [33] was the earliest method to propose full attention sequence parallelism, but it

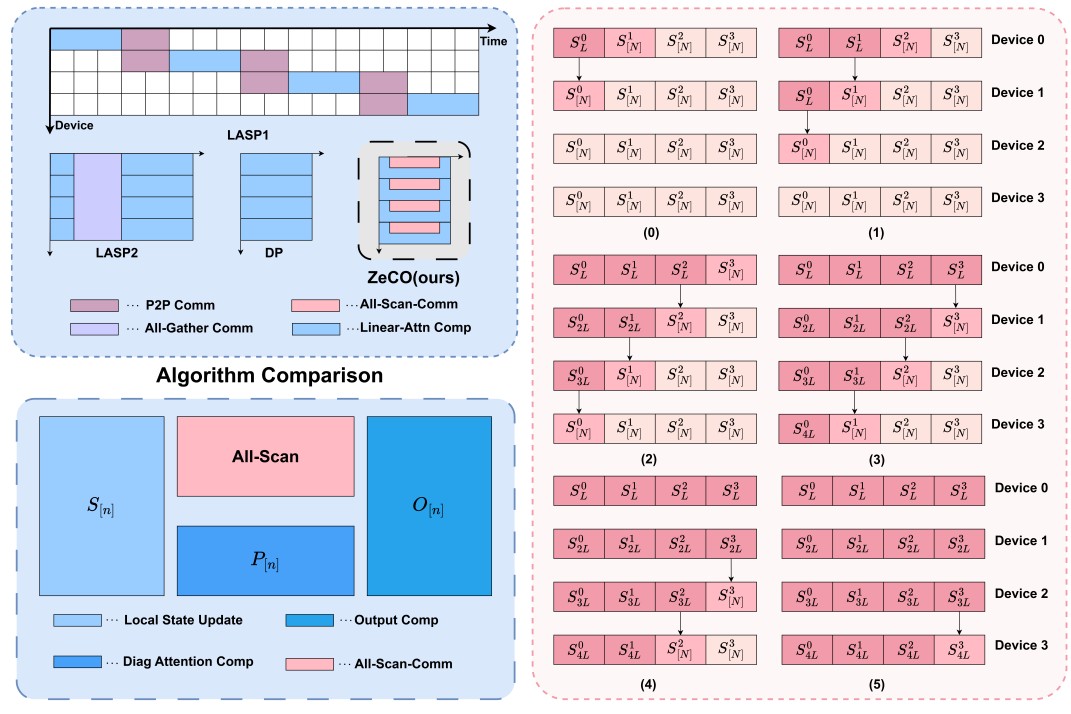

Figure 2: Illustration of ZeCO. ZeCO highlights its strengths in three dimensions: (1) Parallel Scalability: achieving efficiency comparable to DP (sub-figure: Algorithm Comparison); (2) Operator-Level Computation: enabling overlap of communication and local computation for maximal resource utilization (sub-figure: ZeCO Computation); and (3) Communication Pattern: utilizing a customized pipelined All-Scan Communication pattern to substantially reduce inter-device synchronization delays. (sub-figure: All-Scan Communication)

does not leverage the I/O-efficient optimizations of self-attention [34, 35, 36, 37], which limits its applicability. Nevertheless, sequence parallelism for full attention is fundamentally constrained by the self-attention algorithm itself: even disregarding communication, the computational cost becomes prohibitively expensive for ultra-long sequences due to the algorithm's inherent complexity.

## 3  Method

### 3.1  ZeCO Sequence Parallel Methods and Communication Requirements

Let each device be assigned a sequence of length $L$, which is partitioned into $N = L/C$. After projection of input $X \in \mathbb{R}^{L \times d}$, partitioned into $N = L/C$, we get non-overlapping chunks $\mathbf{Q}_{[n]}, \mathbf{K}_{[n]}, \mathbf{G}_{[n]}$ for $n \in N$. We use $\mathbf{S}_i, i \in PL$ to denote the Global states, and $\mathbf{S}_{[\mathbf{n}]}, n \in N$ denote local chunk states on each device.

**Local State Computation.** According to Equation (2). Within each device, we sequentially compute the local states starting from the initial state $\mathbf{S}_{[0]} = \mathbf{0}$:

$$\mathbf{S}_{[n]} = \left(\boldsymbol{\gamma}_{[n]}^\top \mathbf{1}\right) \odot \mathbf{S}_{[n-1]} + \tilde{\mathbf{K}}_{[n]}^\top \mathbf{V}_{[n]}, \quad \text{for} \quad n = 1, \dots, N. \tag{4}$$

Compared to GLA, we additionally maintain the cumulative decay vector $\tilde{\boldsymbol{\gamma}}_{[n]}$, which saves the total multiplicative decay from the first to the $n$-th chunk:

$$\tilde{\boldsymbol{\gamma}}_{[n]} = \prod_{i=0}^{n} \boldsymbol{\gamma}_{[i]}, \quad \text{for} \quad n = 1, \dots, N. \tag{5}$$

At the end of this local recurrence, we obtain a list of local chunk states $\{\mathbf{S}_{[0]}, \mathbf{S}_{[1]} \dots \mathbf{S}_{[N]}\}$

**Global State Update.** The recurrence for the global state is defined as Equation (1). To obtain the series of global states of the device $p$, the device $p$ must get the last global state $S_{(p-1)L}$ from device

---

**Algorithm 1** Forward pass for ZeCO with All-Scan comunication

---

1: **Note:** The Highlighted part represents the SP adaptation of GLA Algorithm by ZeCO algorithm, and the lower cost of the red part represents the lower extra cost of SP.

2: **Input:** $\mathbf{Q}, \mathbf{K}, \in \mathbb{R}^{L \times d_k}, \mathbf{V} \in \mathbb{R}^{L \times d_v}, \mathbf{G} = [\boldsymbol{\alpha}_1 ... \boldsymbol{\alpha}_L] \in \mathbb{R}^{L \times d_k}$, chunk size $C$, num_device $P$, device_rank $p \in \{0, 1, \ldots, P-1\}$

3: Divide $\mathbf{Q}, \mathbf{K}, \mathbf{G}$ into $N = \frac{L}{C}$ blocks $\{\mathbf{Q}_{[1]} ... \mathbf{Q}_{[N]}\}, \{\mathbf{K}_{[1]} ... \mathbf{K}_{[N]}\}, \{\mathbf{G}_{[1]} ... \mathbf{G}_{[N]}\}$ of size $C \times d_k$ each. Divide $\mathbf{V}$ into $N$ blocks $\{\mathbf{V}_{[1]} ... \mathbf{V}_{[N]}\}$ of size $C \times d_v$ each.

4: Initialize $\mathbf{S} = \mathbf{0} \in \mathbb{R}^{d_k \times d_v}, \tilde{\gamma} = \mathbf{1} \in \mathbb{R}^{d_k}$ on SRAM

5: Write $\tilde{\gamma}, \mathbf{S}$ to HBM as $\tilde{\gamma}_{[\mathbf{0}]}, \mathbf{S}_{[\mathbf{0}]}$.

6: **for** $n \leftarrow 0, N$ **do**

7:      Load $\mathbf{K}_{[n]}, \mathbf{G}_{[n]}, \mathbf{V}_{[n]}$ from HBM to SRAM.

8:      On chip, compute $\boldsymbol{\gamma}_{[n]} \in \mathbb{R}^{d_k}, \boldsymbol{\Gamma}_{[n]} \in \mathbb{R}^{C \times d_k}$ and $\bar{\mathbf{K}}_{[n]} = \mathbf{K}_{[n]} \odot \boldsymbol{\Gamma}_{[n]}, \tilde{\gamma} = \tilde{\gamma} \odot \boldsymbol{\gamma}_{[n]}$

9:      Write $\tilde{\gamma}$ to HBM as $\tilde{\gamma}_{[n]}$.

10:      On chip, compute $\mathbf{S} = \left(\boldsymbol{\gamma}_{[n]}^\top \mathbf{1}\right) \odot \mathbf{S} + \tilde{\mathbf{K}}_{[n]}^\top \mathbf{V}_{[n]}$.

11:      Write $\mathbf{S}$ to HBM as $\mathbf{S}_{[n]}$.

12: **end for**

13: **In parallel do:**

14: **parallel stream 1**:

15: $\mathbf{S}_{(p-1)L}, \mathbf{S}_{pL} \leftarrow$ All-Scan$(\mathbf{S}_{[\mathbf{N}]}, \tilde{\gamma}_{[N]})$

16: **parallel stream 2**:

17: **parfor** $n \leftarrow 1, N$ **do**

18:      Load $\mathbf{Q}_{[n]}, \mathbf{K}_{[n]}, \mathbf{G}_{[n]} \in \mathbb{R}^{C \times d_k}$ from HBM to SRAM.

19:      On chip, construct causal mask $\mathbf{M} \in \mathbb{R}^{C \times C}$

20:      On chip, compute $\boldsymbol{\Lambda}_{[n]}, \in \mathbb{R}^{C \times d_k}, \tilde{\mathbf{Q}}_{[n]} = \mathbf{Q}_{[n]} \odot \boldsymbol{\Lambda}_{[n]}, \bar{\mathbf{K}}_{[n]} = \mathbf{K}_{[n]} / \boldsymbol{\Lambda}_{[n]}$

21:      On chip, compute $\mathbf{P} = (\tilde{\mathbf{Q}}_{[n]} \bar{\mathbf{K}}_{[n]}^\top) \odot \mathbf{M} \in \mathbb{R}^{C \times C}$

22:      Write $\mathbf{P}$ as $\mathbf{P}_{[\mathbf{i}]}$ to HBM.

23: **end parfor**

24: **stream barrier**

25: **for** $n \leftarrow 1, N$ **do**

26:      Load $\mathbf{Q}_{[n]}, \mathbf{G}_{[n]}, \mathbf{V}_{[n]}, $$\mathbf{S}_{(p-1)L}$$, \mathbf{S}_{[n]}, \tilde{\gamma}_{[n-1]}, \mathbf{P}$ from HBM to SRAM.

27:      On chip, compute $\boldsymbol{\Lambda}_{[n]}$

28:      On chip, compute $\tilde{\mathbf{Q}}_{[n]} = \mathbf{Q}_{[n]} \odot \boldsymbol{\Lambda}_{[n]}$

29:      On chip, compute $\mathbf{O}_{[n]}^{\text{inter}} = \tilde{\mathbf{Q}}_{[n]}(\mathbf{S}_{[n-1]} + $$(\tilde{\gamma}_{[\mathbf{n-1}]}^\top \mathbf{1}) \odot \mathbf{S}_{(p-1)L}$$), \mathbf{O}^{\text{intra}} = \mathbf{P}\mathbf{V}_{[n]} \in \mathbb{R}^{C \times d_v}$

30:      On chip, compute $\mathbf{O}_{[n]} = \mathbf{O}^{\text{inter}} + \mathbf{O}^{\text{intra}}$

31:      Store $\mathbf{O}_{[n]}$ to HBM.

32: **end for**

33: **return** $\mathbf{O} = \{\mathbf{O}_{[1]} ... \mathbf{O}_{[N]}\}, \mathbf{S} = \{$$\mathbf{S}_{(p-1)L}$$, \mathbf{S}_{[1]} ... \mathbf{S}_{[N]}, $$\mathbf{S}_{pL}$$\}$.

---

$p - 1$. Then we can update the global State of the current device by applying:

$$\mathbf{S}_{(p-1)L+nC} = (\tilde{\gamma}_{[n]}^\top \mathbf{1}) \odot \mathbf{S}_{(p-1)L} + \mathbf{S}_{[n]}. \tag{6}$$

We give a proof of the above global update computation in Appendix A.1. Since updating each local state $\mathbf{S}_{[n]}$ to its global state $\mathbf{S}_{(p-1)L+nC}$ is independent across chunks, we can first update $\mathbf{S}_{[N]}$ to the last global state $\mathbf{S}_{pL}$ of device $p$, and send it to enable the global state update in $p + 1$ device.

To fulfill this communication requirement, we propose the All-Scan Collective Communication operator in Section 3.2. All-Scan Communication is overlapped with the local computations that do not depend on communication; in practice, we compute the diagonal attention scores simultaneously as shown in Figure 2. All-Scan allows ZeCO to parallelize both inter-device communication and intra-device computation. In implementation, ZeCO rearranges the standard form of GLA with minimal extra computation and I/O cost, so as to achieve efficient sequential parallel training Algorithm 1.

### 3.2 All-Scan Collective Communication

To convert local chunk states in each device into globally consistent values, each device $p$ requires the final state $\mathbf{S}_{(p-1)L}$ from its predecessor. It presents a dependency chain between devices, which

---

**Algorithm 2** All-Scan Algorithm

---

1: **Input:** num_device $P$, device_rank $p \in [P]$, Local State $\mathbf{S}_{\text{local}}$, factor $\tilde{\gamma}$, the direction tag **DIR**
2: **if DIR == FWD then**
3:     $send\_rank = p + 1$, $recv\ rank = p - 1$ for device $p$, $start = 0$, $last = P - 1$
4: **else if DIR == BWD then**
5:     compute $send\_rank = p - 1$, $recv\_rank = p + 1$ for device $p$, $start = P - 1$, $last = 0$
6: **end if**                                                       ▷ Single Direction Communication
7: Initialize the send state as $\mathbf{S}_{\text{send}}$,
8: **if** $p$ is not 0 **then**
9:     Receive state from $recv\_rank$ as $\mathbf{S}_{recv}$,
10: **end if**
11: Slice $\mathbf{S}_{\text{recv}}, \mathbf{S}_{\text{send}}, \mathbf{S}_{\text{local}}, \tilde{\gamma}$ alone the first dimension in $K$ blocks.
12: **for** $k \leftarrow 0, K - 1$ **do**
13:     **if** $p$ is $start$ **then**
14:         Send $\mathbf{S}_{\text{local}}$ to $recv\_rank$
15:     **else if** $p$ is not 0 **then**
16:         $\mathbf{S}_{\text{send}}^k = \mathbf{S}_{\text{local}}^k + (\tilde{\gamma}_{[N]}^{(k)} \mathbf{1}) \times \mathbf{S}_{\text{recv}}^k$
17:         Send $\mathbf{S}_{\text{local}}$ to $recv\_rank$
18:     **else if** $p$ is $last$ **then**
19:         $\mathbf{S}_{\text{send}}^k = \mathbf{S}_{\text{local}}^k + (\tilde{\gamma}_{[N]}^{(k)} \mathbf{1}) \times \mathbf{S}_{\text{recv}}^k$
20:     **end if**
21: **end for**
22: **return** $\mathbf{S}_{\text{recv}}, \mathbf{S}_{\text{send}}$

---

will cause a communication latency related to the number of devices. To address this efficiently, we propose an **All-Scan Collective Communication** strategy to receive, update, and send. Specifically, All-Scan splits large state tensors into smaller segments that can be sequentially transmitted and processed.

**Pipelined State Scan.**  Rather than receive the full state $\mathbf{S}_{(p-1)L}$, we partition it along the $d_k$ dimension into $K$ contiguous blocks to send from device $p - 1$:

$$\mathbf{S}_{(p-1)L} = \left[ \mathbf{S}_{(p-1)L}^{(1)}, \mathbf{S}_{(p-1)L}^{(2)}, \ldots, \mathbf{S}_{(p-1)L}^{(K)} \right], \quad \mathbf{S}_{(p-1)L}^{(k)} \in \mathbb{R}^{\frac{d_k}{K} \times d_v}. \tag{7}$$

Correspondingly, the decay factor is split into aligned segments $\tilde{\gamma}_{[N]}^{(j)} \in \mathbb{R}^{1 \times \frac{d_k}{K}}$. Each block of state is transmitted pipelined from $p - 1$ to $p$, and immediately applies the update and send:

$$\mathbf{S}_{pL}^{(k)} = (\tilde{\gamma}_{[N]}^{(k)\top} \mathbf{1}) \odot \mathbf{S}_{(p-1)L}^{(k)} + \mathbf{S}_{[N]}^{(k)} \quad \text{for} \quad k = 0, ..., K \tag{8}$$

This design enables device $p + 1$ to begin updating its last global state $\mathbf{S}_{(p+1)L}$ in All-Scan as soon as it receives the first block of $\mathbf{S}_{pL}$, as shown in Algorithm 2. As a Communication Primitive, All-Scan could run independently with other CUDA stream, achieves fine-grained communication-computation overlap, maximizing device utilization and throughput in long-context training.

### 3.3  Efficiency Analysis

We now formally analyze the time cost of sequence parallelism strategies and show that using the All-Scan collective communication algorithm yields the most efficient design for linear attention SP under our analytical framework. We identify two key conditions that characterize such efficiency and demonstrate that ZeCO satisfies both.

1. **Zero Communication Overhead**: Each device transmits and receives only the minimal essential size of information (data). No redundant communication.

2. **Low Extra Cost**: Communication is overlapped with other computations as much as possible to minimize idle computational resources. Furthermore, the additional computation and I/O overhead introduced by SP is reduced to a minimum.

**Zero Communication Overhead**  Let $S \in \mathbb{R}^{d_k \times d_v}$ denote the accumulated state tensor. According to linear attention output Equation (3), this state represents the minimal information that must be communicated between chunks.

For a sequence distributed across $P$ devices, any SP algorithm must communicate at least the state information across device boundaries. Let $V_{\text{ZeCO}}^{(p)}$ denote the communication volume of the $p$-th device in ZeCO (All-Scan) Each device sends a last global state to the next device exactly once, resulting in:

$$V_{\text{ZeCO}}^{(p)} = |S| = d_k \times d_v. \tag{9}$$

This represents the theoretical lower bound. However, existing approaches such as LASP-2 rely on all-gather operations, receive local states from all the other $P - 1$ devices, resulting in a communication volume of $(P - 1) \times d_k \times d_v$, which increases with the number of devices.

Which grows linearly with $P$. As shown in Figure 3, ZeCO achieves the minimal communication volume possible for the SP scenario.

**Idealized SP strategy**  Let $T_{\text{SP}}^P(L)$ denote the total runtime for processing a sequence of length $L$ using $P$ devices with sequence parallelism, and let $T_{\text{SP}}^1(PL)$ represent the runtime for processing a total sequence of length $PL$ on a single device. Let $T_{\text{ideal-SP}}^P(PL)$ denote the runtime under ideal conditions, assuming perfect parallelism with zero additional overhead. For an ideal SP, the following properties should be satisfied:

$$T_{\text{ideal-sp}}^P(PL) = T_{\text{ideal-SP}}^1(L) = \frac{T_{\text{ideal-SP}}^1(PL)}{P}. \tag{10}$$

The implication of Equation (10) is that, in the ideal case, the throughput of sequence parallelism should scale linearly with the number of devices (i.e., the processing time is inversely proportional to the throughput). In practice, however, sequence parallelism introduces additional overhead. Therefore, we next analyze the latency under practical scenarios.

In practice, the prerequisite communication latency, computation, and I/O for transferring and synchronizing data between devices introduce additional latency. For ZeCO (and other sequence parallelism), the relation becomes ideal sequence parallelism cost + extra cost, which can be formularized as follow:

$$T_{\text{SP}}^P(PL) = T_{\text{ideal-SP}}^P(PL) + \mathbf{T}_{\text{extra\_comp\&I/O}} + (\mathbf{T}_{\text{All\_Scan}} - T_{\text{overlaped\_comp}}). \tag{11}$$

For ZeCO, the first two components are independent of communication. The last two represent communication latency, accounting for the portion of the All-Scan operator that cannot be overlapped by local diagonal attention (we suppose the worst-case scenario). This can also be viewed as the gap relative to the ideal SP. From Equation (11), we can see that $T_{\text{ideal-SP}}^P(PL)$ and $T_{\text{overlaped\_comp}}$ are inherent to the algorithm. Therefore, the key question in SP is to what extent the additional time cost of linear attention, namely $\mathbf{T}_{\text{extra\_comp\&I/O}}$ and $\mathbf{T}_{\text{All\_Scan}}$, can be reduced. The following presents how ZeCO achieves its efficiency through an analysis of the All-Scan communication strategy, and proves that the additional computation and I/O overhead is negligible.

For ZeCO, the first two components are independent of communication, while the last two correspond to communication latency—specifically, the portion of the All-Scan operation that cannot be fully overlapped by local diagonal attention (we consider the worst-case scenario). This latency can also be viewed as the residual gap relative to the ideal SP case. From Equation (11), $T_{\text{ideal-SP}}^P(PL)$ and $T_{\text{overlaped\_comp}}$ are intrinsic to the algorithm design. Hence, the central question in SP is to what extent the additional costs of linear attention, namely $\mathbf{T}_{\text{extra\_comp\&I/O}}$ and $\mathbf{T}_{\text{All\_Scan}}$, can be minimized. In the following, we show that ZeCO achieves near-optimal efficiency by analyzing the All-Scan communication behavior, and we further demonstrate that extra_computation&I/O constitutes a negligible portion of the overall system cost.

In All-scan, we partition $S$ into $K$ blocks, each of size $\mathbb{R}^{\frac{d_k}{K} \times d_v}$, and transmit these blocks in a pipelined fashion as shown in Figure 2. By partitioning the state $S$ into $K$ blocks and updating them in a pipeline, the effective communication latency could be computed as:

$$\mathbf{T}_{\text{All\_Scan}} = \tau(d_k \times d_v) + \frac{(P - 1)\tau(d_k \times d_v)}{K}, \tag{12}$$

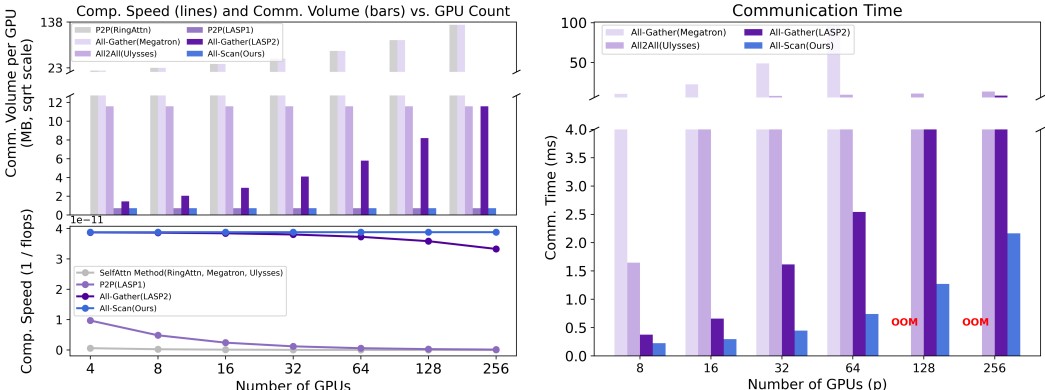

Figure 3: ZeCO has the lowest communication time while satisfying the lowest communication volume. The left two figures show the theoretical values of the algorithm calculation speed and communication volume, and the right figure shows the actual communication time.

where $\tau(\cdot)$ represents the time required to communicate a tensor of the given size. Equation12 shows the two components of the cost $\mathbf{T}_{\text{All\_Scan}}$. The first term represents the overhead that can be parallelized by the pipelined approach, which is necessary and corresponds to the minimum communication requirement. The second term accounts for the overhead at the boundaries. As $K$ increases, the boundary overhead decrease, and the degree of overlap improves. Consequently, when $K$ becomes sufficiently large, the boundary overhead approaches zero. Thus, ZeCO with All-Scan achieves the minimal time cost of communication.

The term $\mathbf{T}_{\text{extra\_comp\&I/O}}$ consists of two parts: a small number of additional floating-point operations, and HBM load and store operations for a few auxiliary tensors. In Algorithm 1, the load and store operations for $\tilde{\gamma}_{[n]}$ in lines 9 and 26 are vector, constituting only $\frac{1}{d_v}$ of the state tensor. The required additional state can be reused $N$ times, incurring just a $\frac{1}{N}$ overhead. For a sequence length of 8192 and a chunk size of 64 ($N = 128$), which is comparable to typical $d_v$, the added overhead is less than 1%. The cost of element-wise multiplications is negligible. Hence, $\mathbf{T}_{\text{extra\_comp\&I/O}}$ can be safely ignored in practice. It proved that the time of ZeCO Equation (11) should be:

$$T_{\text{ZeCO}}^{P}(PL) = T_{\text{ideal-SP}}^{1}(L) - T_{\text{overlaped\_comp}} + \tau(d_k \times d_v) + \epsilon \tag{13}$$

$$\approx T_{\text{ideal-SP}}^{1}(L) - T_{\text{overlaped\_comp}} + \tau(d_k \times d_v), \tag{14}$$

where $\epsilon$ represents a negligible computation and I/O cost.

In contrast, existing methods like LASP have a strictly serial dependency across devices, resulting in (We assume that the $\mathbf{T}_{\text{extra\_comp\&I/O}}$ term in other methods can also be optimized to a negligible level. Even so, these methods remain suboptimal. ):

$$T_{\text{LASP}}^{P}(PL) = P \times (T_{\text{ideal-SP}}^{1}(L) + \tau(d_k \times d_v)) > T_{\text{ZeCO}}^{P}(PL). \tag{15}$$

While LASP-2 improves on LASP with parallel computation, but suffers from higher communication cost:

$$T_{\text{LASP-2}}^{P}(PL) = T_{\text{ideal-SP}}^{1}(L) + P \times \tau(d_k \times d_v) > T_{\text{ZeCO}}^{P}(PL). \tag{16}$$

Thus, ZeCO effectively eliminates redundant communication overhead and establishes an efficient sequence parallelism strategy with minimal extra cost. This efficiency directly translates into superior performance.In Figure 3, we show the theoretical values of communication cost and computational overhead, and the actual values of communication time for different SP algorithms. We also present a unified communication and runtime analysis of existing SP algorithm in Appendix A.2.

## 4 Experiments

We evaluate the efficiency and scalability of the proposed ZeCO SP Algorithm and All-Scan Communication Operator on 1B-GLA models. Our assessment focuses on two aspects: (1) Communication

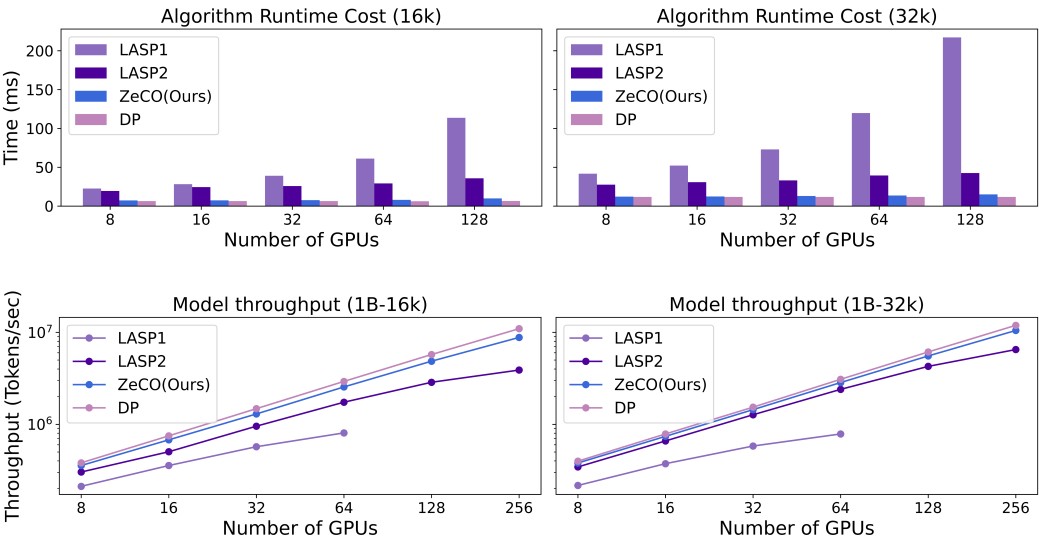

Figure 4: Scalability evaluation of LASP on SP operator runtime(top half) and Scalability evaluation of LASP on Throughput(bottom half). In the comparison test of 16k and 32k sequence length per GPU, ZeCO algorithm shows the same stable time as the DP algorithm. In both 16k and 32k, ZeCO exhibits a linear scaling curve of throughput growth approach to DP, while the other methods degenerate.

speed of different Collective Communication Operators; (2) The Algorithm-level and model-level scalability under increasing GPU count.

All experiments are conducted on a GPU cluster equipped with 256×H100 80GB GPUs. Model is trained in Lingua [38], a PyTorch-based distributed training. We implement the All-Scan communication algorithm using the *Triton-Distributed* framework, which integrates OpenSHMEM into the Triton compiler to enable distributed communication within operator implementations [39, 40]. To ensure a fair comparison with baseline sequence parallelism (SP) methods such as LASP1 [17] and LASP2 [18], we adapt the chunk-wise gated linear attention operator from the Flash Linear Attention [41] repository for our implementation. The complete experimental setup and data are provided in the Appendix A.3.

## 4.1 Communication Speed

In this experiment, we evaluate the communication Runtime of different communication operators under their own communication workload sufficient for correct training. Experiments are conducted with $P$ from 8 to 256 GPUs, and each GPU is assigned 8K sequence length.

We warm up each communication kernel for 5 rounds, then report the average over 50 runs. More details of communication workloads and protocol differences are discussed in Appendix A.2.

As shown in Figure 3, memory-out occurred in the experiments of 128 GPUs and 256 GPUs All-gather (Megatron). For other methods, it should be noted that, for presentation purposes, the upper half of the Y-axis represents the rendering results after taking the log scale. All-Scan significantly outperforms other methods in different scales of clusters. Notably, on 256 GPUs, All-Gather (LASP2) is $4\times$ slower than All-Scan.

## 4.2 SP Algorithm Runtime and Model Throughput

Next, we evaluate both micro-level (algorithm) and macro-level (model training) performance for Linear Attention SP methods, including LASP1, LASP2, and ZeCO.

**SP Algorithm Runtime** We measure the forward and backward pass time of each SP operator under the same setting ($L = 16K$ *or* 32K, $H = 16$) and compare it against the ideal case of a DP operator. The time of the DP operator serves as the theoretical lower bound.

Figure 4 demonstrates that in the 128 GPUs experiment (2M and 4M sequence length), ZeCO is only 3 ms slower than the theoretical lower bound for a single forward and backward pass, which satisfies our analysis in Section 3.3 and demonstrates the efficiency of our algorithm.

**Model Throughput**   We experimented with 1B-GLA models with different sequence parallel methods to test the training throughput, under the same setting ($L = 16\text{K } or\text{ } 32\text{K}$, $H = 16$) and compare it against the ideal case of a Model that uses DP. The throughput of the GLA model uses DP in training, serves as the theoretical upper bound.

For the result shown in Figure 4, as the number of GPUs increases, ZeCO achieves a linear increase in total throughput, which meets the original intention of sequence parallelism, while other methods experience a serious degradation.

## 5   Conclusion and Future Works

In this work, we propose ZeCO sequence parallelism for linear attention, achieving SOTA for both theoretical and empirical results. More importantly, our method fully unleashes the algorithmic efficiency of linear models and, for the first time, enables near-linear throughput scaling for sequence parallelism. At the system level, our approach introduces the novel All-Scan collective communication primitive, which not only underpins the efficiency of ZeCO but also provides a foundational innovation for advancing distributed computing in the linear model community.

In the future, we plan to pursue three main directions. First, we will further improve the algorithmic implementation of the All-Scan collective communication primitive. For example, tree-like implementation. Second, we aim to generalize the sequence parallelism algorithm for linear attention beyond diagonal decay, extending it to support various forms, including matrix transform structures. Third, we will investigate efficient parallel topologies for sequence parallelism in large-scale models.

## Acknowledgments and Disclosure of Funding

This work was supported by the Research Grants Council of the Hong Kong SAR (Grant No. C5052-23G, PolyU15217424, PolyU25216423), and , The Hong Kong Polytechnic University (Project IDs: P0043563).

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

# NeurIPS Paper Checklist

The checklist is designed to encourage best practices for responsible machine learning research, addressing issues of reproducibility, transparency, research ethics, and societal impact. Do not remove the checklist: **The papers not including the checklist will be desk rejected.** The checklist should follow the references and follow the (optional) supplemental material. The checklist does NOT count towards the page limit.

Please read the checklist guidelines carefully for information on how to answer these questions. For each question in the checklist:

- You should answer [Yes] , [No] , or [NA] .

- [NA] means either that the question is Not Applicable for that particular paper or the relevant information is Not Available.

- Please provide a short (1–2 sentence) justification right after your answer (even for NA).

**The checklist answers are an integral part of your paper submission.** They are visible to the reviewers, area chairs, senior area chairs, and ethics reviewers. You will be asked to also include it (after eventual revisions) with the final version of your paper, and its final version will be published with the paper.

The reviewers of your paper will be asked to use the checklist as one of the factors in their evaluation. While "[Yes] " is generally preferable to "[No] ", it is perfectly acceptable to answer "[No] " provided a proper justification is given (e.g., "error bars are not reported because it would be too computationally expensive" or "we were unable to find the license for the dataset we used"). In general, answering "[No] " or "[NA] " is not grounds for rejection. While the questions are phrased in a binary way, we acknowledge that the true answer is often more nuanced, so please just use your best judgment and write a justification to elaborate. All supporting evidence can appear either in the main paper or the supplemental material, provided in appendix. If you answer [Yes] to a question, in the justification please point to the section(s) where related material for the question can be found.

IMPORTANT, please:

- **Delete this instruction block, but keep the section heading "NeurIPS Paper Checklist",**

- **Keep the checklist subsection headings, questions/answers and guidelines below.**

- **Do not modify the questions and only use the provided macros for your answers**.


# A  Appendix: Supplementary Proof and Experimental Data

## A.1  Global Chunk Update Proof

We prove the correctness of the global correction formula used in Equation (6) of the main text, which expresses the global state at position $(p-1)L + nC$ as:

$$\mathbf{S}_{(p-1)L+nC} = (\tilde{\boldsymbol{\gamma}}_{[n]}\mathbf{1}) \odot \mathbf{S}_{(p-1)L} + \mathbf{S}_{[n]}. \tag{17}$$

We begin from the chunkwise recurrence of the Gated Linear Attention (GLA) state update within each device. For any chunk $n$, the recurrence is:

$$\mathbf{S}_{[n]} = \left(\boldsymbol{\gamma}_{[n]}^{\top}\mathbf{1}\right) \odot \mathbf{S}_{[n-1]} + \tilde{\mathbf{K}}_{[n]}^{\top}\mathbf{V}_{[n]}, \tag{18}$$

with initial state $\mathbf{S}_{[0]} = \mathbf{0}$. Unfolding the recurrence, we obtain the closed-form expression of the final local state $\mathbf{S}_{[n]}$:

$$\mathbf{S}_{[n]} = \sum_{i=1}^{n}\left(\prod_{j=i+1}^{n}\boldsymbol{\gamma}_{[j]}^{\top}\mathbf{1}\right) \odot \left(\tilde{\mathbf{K}}_{[i]}^{\top}\mathbf{V}_{[i]}\right) + \left(\prod_{j=1}^{n}\boldsymbol{\gamma}_{[j]}^{\top}\mathbf{1}\right) \odot \mathbf{S}_{[0]}. \tag{19}$$

Equation 19 represents the result of local computation, it captures the final local state obtained by starting from a zero initial state and considering only the local contribution within the current chunk $n$. The key observation is that the second term $\left(\prod_{j=1}^{n}\gamma_{[j]}^{\top}\right) \odot S_{[0]}$ vanishes due to the initial condition $S_{[0]} = \mathbf{0}$, making $S_{[n]}$ completely determined by local information. We now demonstrate the linear decomposition property of global state updates. The key insight is that when non-zero initial state exists, the final global state can be decomposed into two independent linear contributions: attenuated propagation of global computation and the current chunk's local contribution. Now suppose we instead perform the same recurrence starting from a **non-zero** initial state $\mathbf{S}_{(p-1)L}$, which is the final global state of the previous device. The updated state at global index $(p-1)L + nC$ becomes:

$$\mathbf{S}_{(p-1)L+nC} = \sum_{i=1}^{n}\left(\prod_{j=i+1}^{n}\boldsymbol{\gamma}_{[j]}^{\top}\mathbf{1}\right) \odot \left(\tilde{\mathbf{K}}_{[i]}^{\top}\mathbf{V}_{[i]}\right) + \left(\prod_{j=1}^{n}\boldsymbol{\gamma}_{[j]}^{\top}\mathbf{1}\right) \odot (\mathbf{0} + \mathbf{S}_{(p-1)L}) \tag{20}$$

$$= (\tilde{\boldsymbol{\gamma}}_{[n]}\mathbf{1}) \odot \mathbf{S}_{(p-1)L} + \mathbf{S}_{[n]}. \tag{21}$$

This linear property allows local computation $\mathbf{S}_{[n]}$ stores only the residual contribution from chunk $n$, and multiplying the incoming global state $\mathbf{S}_{(p-1)L}$ by the cumulative decay $\tilde{\gamma}_{[n]}$ precisely reconstructs the full global state.

## A.2  Unified Analysis of Sequence Parallel Methods

In this section, we use multi-head attention with head $H = 32$, $d_k = d_v = \frac{D}{H} = e$. We present a unified analysis of several representative sequence parallel (SP) methods across both full attention and linear attention models. Specifically, we compare them from the following three perspectives:

- **Communication Volume**: The total amount of data transferred per device during SP execution.

- **Computation Cost**: The total computation time to process a sequence of length $PL$ in parallel.

- **Additional Computation Overhead**: The extra operations introduced due to SP-specific logic.

**Full Attention Models.**

- **Ulysses**: Uses All-to-All communication to exchange Q, K, V, and Output tensors. Communication volume is $4LD$ per device. Due to full attention's quadratic complexity, the computation cost is $L^2DP$.

- **Megatron CP**: Utilizes All-Gather to collect Q and K from all devices. Communication volume is $2PLD$. Computation cost is the same as Ulysses, $L^2DP$.

**Linear Attention Models.** In linear attention, since each device processes a sequence of length $L$, the inherent computation per device is $LDe$ (here we ignore the additional lower-order terms introduced by the chunk-wise algorithm), which is independent of $P$.

- **LASP-1**: Employs serial P2P communication. Each device transmits a single state tensor $S \in \mathbb{R}^{H \times e \times e}$, with communication volume $De$. However, the devices execute sequentially, resulting in an equivalent time overhead (including both communication and computation) as if each device performed $P$ times the workload.

- **LASP-2**: Uses All-Gather to collect all intermediate state tensors across devices. Each device processes all $P$ global states, leading to $PDe$ communication volume and additional computation cost of $\log(P)De + NDe$ for sum reduction and state updates.

- **ZeCO (Ours)**: Implements pipelined communication via All-Scan. Each device sends/receives only one state $S$, with communication volume $De$. It additionally maintains $N$ cumulative decay vectors $\tilde{\gamma}$ and updates $N$ intermediate states using global recurrence. So, here is an extra computation cost $NDe + Nd$.

In conclusion, for sequence parallelism with full attention, both the computation cost and the number of device parameters are strongly dependent on the number of devices $P$, which becomes a major efficiency bottleneck. In the case of linear attention, although both the communication and computation costs of LASP-1 and LASP-2 scale with $P$, the computation cost constitutes only a small fraction of the total overhead. As a result, the communication cost's dependence on $P$ becomes the primary bottleneck. In contrast, our ZeCO algorithm achieves both communication and computation costs that are independent of the number of devices $P$.

Table 1: Comparison of Sequence Parallel Methods: Communication Volume and Computation Cost (For LASP-1, we consider the sequential execution order)

| Method | Communication Volume | Computation Cost |
|---|---|---|
| Ulysses (Full) | $4LD$ | $L^2DP$ |
| Megatron CP (Full) | $2PLD$ | $L^2DP$ |
| LASP-1 (Linear) | $PDe$ | $PLDe$ |
| LASP-2 (Linear) | $PDe$ | $LDe + \log(P)De + NDe$ |
| ZeCO (Ours, Linear) | $De$ | $LDe + NDe + Nd$ |

## A.3 Experimental Setting and Supplementary Data

In experiment Section 4.1, $H$ is 32, the tensor size of each chunk of segmentation is 16384, the hidden dimension $d$ is 4096, and sequence length per device $L$ is 8192. The experimental setup with 5 rounds of warm-up and 50 rounds of experiment was averaged, see in Table 2.

In experiment Section 4.2, In the experiment of algorithm run time, we test the GLA-attention algorithm equipped with different SP methods, record the time of 1 iteration of FWD and BWD. $H$ is 16, the tensor size of each chunk of segmentation is 16384, the hidden dimension $d$ is 2048, and sequence length per device $L$ is 16384 and 32768. The experimental setup with 5 rounds of warm-up and reported the average of 50 rounds of experiment, see in Table 3, Table 4. In the experiment of Model throughput, we test the GLA-1B Model equipped with different SP methods, and record the throughput in the training stage. $H$ is 32, the tensor size of each chunk of segmentation is 16384, the number of model layers is 20, the hidden dimension $d$ is 2048, and the sequence length per device $L$ is 16384 and 32768. The experimental setup with 5 rounds of warm-up reported the average of 100 steps of the experiment, see in Table 5, Table 6.

## A.4 Backward pass for ZeCO with All-Scan comunication

In the backward propagation of the ZeCO algorithm, most of the process is similar to the forward propagation. It is important to note the difference in notation here: $\tilde{\gamma}_{[n]}$ denotes the decay factor for the reverse cumulative product. Furthermore, in the official implementation of gated linear attention, $\mathbf{S}_{[n]}$ needs to be recomputed during the backward pass. However, since the global initial state has already been obtained during the forward pass, there is no need for all-scan communication when recomputing $\mathbf{S}_{[n]}$.

**Algorithm 3** Backward pass for ZeCO with All-Scan comunication

---

**Input:** $\mathbf{Q}, \mathbf{K}, \mathbf{G} \in \mathbb{R}^{L \times d_k}, \mathbf{V}, \mathbf{dO} \in \mathbb{R}^{L \times d_v}$, chunk size $C$, num_device $P$, device_rank $p \in \{0, 1, \ldots, P-1\}$

Initialize $\mathbf{dS} = \mathbf{0} \in \mathbb{R}^{d_k \times d_v}$ on SRAM

1: **for** $n \leftarrow N$ **to** $0$ **do**
2:     Load $\mathbf{G}_{[n]} \in \mathbb{R}^{C \times d_k}, \mathbf{Q}_{[n]} \in \mathbb{R}^{C \times d_k}, \mathbf{dO}_{[n]} \in \mathbb{R}^{C \times d_v}$ from HBM to SRAM
3:     On chip, compute $\boldsymbol{\gamma}_{[n]}, \boldsymbol{\Gamma}_{[n]}$ and $\tilde{\mathbf{Q}}_{[n]} = \mathbf{Q}_{[n]} \odot \mathbf{G}_{[n]}, \tilde{\boldsymbol{\gamma}} = \tilde{\boldsymbol{\gamma}} \odot \boldsymbol{\gamma}_{[n]}$
4:     Store $\tilde{\boldsymbol{\gamma}}$ in HBM as $\tilde{\boldsymbol{\gamma}}_{[n]}$
5:     On chip, compute $\mathbf{dS} = (\boldsymbol{\gamma}_{[n]}^{\top} \mathbf{1}) \odot \mathbf{dS} + \tilde{\mathbf{Q}}_{[n]}^{\top} \mathbf{dO}_{[n]}$
6:     Store $\mathbf{dS}$ in HBM as $\mathbf{dS}_{[n]}$
7: **end for**
8: **In parallel do:**
9: **parallel stream 1**:
10: $\mathbf{dS}_{(p-1)L}, \mathbf{dS}_{pL} \leftarrow$ All-Scan$(\mathbf{dS}_{[0]}, \tilde{\boldsymbol{\gamma}}_{[0]})$
11: **parallel stream 2**:
12: Load $\mathbf{S}_{(p-1)L}$ from HBM to SRAM
13: On chip, recompute $\mathbf{S}_{[n]}$ with $\mathbf{S}_{[0]} = \mathbf{S}_{(p-1)L}, n = \{0, 1, 2, \ldots, N-1\}$
14: Store $\{\mathbf{S}_{[n]}, n \in \{0, 1, 2, \ldots, N-1\}\}$
15: **for** $n \leftarrow 1$ **to** $N$ **in parallel do**
16:     Load $\mathbf{Q}_{[n]}, \mathbf{K}_{[n]}, \mathbf{G}_{[n]}, \mathbf{V}_{[n]}, \mathbf{dO}_{[n]}$ from HBM to SRAM
17:     Load $, \in \mathbb{R}^{d_k \times d_v}$, from HBM to SRAM
18:     On chip, construct causal mask $\mathbf{M} \in \mathbb{R}^{B \times B}$
19:     On chip, compute $\boldsymbol{\Lambda}_{[n]}, \boldsymbol{\Gamma}_{[n]} \in \mathbb{R}^{C \times d_k}$
20:     On chip, compute $\tilde{\mathbf{Q}}_{[n]} = \mathbf{Q}_{[n]} \odot \boldsymbol{\Lambda}_{[n]}, \tilde{\mathbf{K}}_{[n]} = \mathbf{K}_{[n]} \odot \boldsymbol{\Gamma}_{[n]}$
21:     On chip, compute $\mathbf{P}_{[n]} = (\tilde{\mathbf{Q}}_{[n]} \tilde{\mathbf{K}}_{[n]}^{\top}) \odot \mathbf{M} \in \mathbb{R}^{C \times C}$
22:     On chip, compute $\mathbf{dP}_{[n]} = (\mathbf{dO}_{[n]} \mathbf{V}_{[n]}^{\top}) \odot \mathbf{M}$
23:     On chip, compute $\mathbf{d\bar{K}}_{[n]} = \tilde{\mathbf{Q}}_{[n]}^{\top} \mathbf{dP}$
24:     On chip, compute $\mathbf{dK}_{[n]} = \mathbf{d\bar{K}}_{[n]} / \boldsymbol{\Lambda}_{[n]}$
25:     On chip, compute $\mathbf{d\tilde{Q}}_{[n]} = \mathbf{dP}\bar{\mathbf{K}}_{[n]}$
26:     On chip, compute $\mathbf{dQ}_{[n]} = \mathbf{d\tilde{Q}}_{[n]} \odot \boldsymbol{\Lambda}_{[n]}$
27:     Store $\mathbf{P}_{[n]}, \mathbf{dQ}_{[n]}, \mathbf{dK}_{[n]}$ in HBM.
28: **end for**
29: **stream barrier**
30: **for** $n \leftarrow 1$ **to** $N$ **in parallel do**
31:     Load $\mathbf{P}_{[n]}, \mathbf{dQ}_{[n]}, \mathbf{dK}_{[n]}, \mathbf{dO}_{[n]}, \mathbf{Q}_{[n]}, \mathbf{K}_{[n]}, \mathbf{G}_{[n]}, \tilde{\boldsymbol{\gamma}}_{[n-1]}, \mathbf{dS}_{pL}, \mathbf{S}_{[n-1]}$, from HBM to SRAM
32:     On chip, compute $\boldsymbol{\Lambda}_{[n]}, \boldsymbol{\Gamma}_{[n]} \in \mathbb{R}^{C \times d_k}$
33:     On chip, compute $\tilde{\mathbf{K}}_{[n]} = \mathbf{K}_{[n]} \odot \boldsymbol{\Gamma}_{[n]}$
34:     On chip, compute $\mathbf{d\tilde{K}}_{[n]} = \mathbf{V}_{[n]}(\mathbf{dS}_{[n-1]}^{\top} + (\tilde{\boldsymbol{\gamma}}_{[n-1]}^{\top} \mathbf{1}) \odot \mathbf{dS}_{pL}^{\top})$
35:     On chip, compute $\mathbf{dK}_{[n]} = \mathbf{dK}_{[n]} + \mathbf{d\tilde{K}}_{[n]} \odot \boldsymbol{\Gamma}_{[n]}$
36:     On chip, compute $\mathbf{d\tilde{Q}}_{[n]} = \mathbf{dO}_{[n]} \mathbf{S}_{[n-1]}^{\top}$
37:     On chip, compute $\mathbf{dQ}_{[n]} = \mathbf{dQ}_{[n]} + \mathbf{d\tilde{Q}}_{[n]} \odot \boldsymbol{\Lambda}_{[n]}$
38:     On chip, compute $\mathbf{dV}_{[n]} = \mathbf{P}_{[n]}^{\top} \mathbf{dO}_{[n]} + \tilde{\mathbf{K}}_{[n]}(\mathbf{dS}_{[n-1]}^{\top} + (\tilde{\boldsymbol{\gamma}}_{[n-1]}^{\top} \mathbf{1}) \odot \mathbf{dS}_{pL}^{\top})$
39:     Store $\mathbf{dK}_{[n]}, \mathbf{dV}_{[n]}$ in HBM
40: **end for**
41: Let $\mathbf{dQ} = \{\mathbf{dQ}_{[1]}, \ldots, \mathbf{dQ}_{[N]}\}, \mathbf{dK} = \{\mathbf{dK}_{[1]}, \ldots, \mathbf{dK}_{[N]}\}, \mathbf{dV} = \{\mathbf{dV}_{[1]}, \ldots, \mathbf{dV}_{[N]}\}$
42: Compute $\mathbf{dA} = \mathbf{Q} \odot \mathbf{dQ} - \mathbf{K} \odot \mathbf{dK}, \mathbf{dG} = \texttt{revcum}(\mathbf{dA})$
43: **return** $\mathbf{dQ}, \mathbf{dK}, \mathbf{dV}, \mathbf{dG}$

---

Table 2: Communication Runtime

| GPU Number | Method | | |
|---|---|---|---|
| | All Gather Linear | All-Scan | All Reduce |
| 8 | 0.37488 | 0.22578 | 0.1375 |
| 16 | 0.65769 | 0.29686 | 0.22803 |
| 32 | 1.61594 | 0.44775 | 0.31073 |
| 64 | 2.54305 | 0.73899 | 0.41486 |
| 128 | 4.35 | 1.27166 | 0.50084 |
| 256 | 8.51388 | 2.16454 | 0.60405 |

Table 3: Algorithm Runtime (16k sequence length)

| GPU Number | Method | | | |
|---|---|---|---|---|
| | LASP1 | LASP2 | ZeCO | GLA (baseline) |
| 8 | 22.59ms | 19.39ms | 7.32ms | 6.39ms |
| 16 | 28.20ms | 24.48ms | 7.45ms | 6.47ms |
| 32 | 39.03ms | 25.72ms | 7.65ms | 6.44ms |
| 64 | 61.18ms | 29.17ms | 8.04ms | 6.12ms |
| 128 | 113.71ms | 35.72ms | 9.88ms | 6.55ms |

Table 4: Algorithm Runtime (32k sequence length)

| GPU Number | Method | | | |
|---|---|---|---|---|
| | LASP1 | LASP2 | ZeCO | GLA (baseline) |
| 8 | 41.64ms | 27.57ms | 12.12ms | 11.76ms |
| 16 | 52.14ms | 30.80ms | 12.41ms | 11.74ms |
| 32 | 72.97ms | 33.08ms | 12.98ms | 11.74ms |
| 64 | 119.79ms | 39.44ms | 13.56ms | 11.79ms |
| 128 | 217.20ms | 42.50ms | 15.06ms | 11.74ms |

Table 5: GLA Model SP Throughput/GPU (tokens/sec) on 1B-16k

| GPU Number | Method | | | |
|---|---|---|---|---|
| | LASP1 | LASP2 | ZeCO | GLA (baseline) |
| 8 | 26428 | 37812 | 44497 | 47594 |
| 16 | 22244 | 31415 | 42328 | 46786 |
| 32 | 17813 | 29802 | 40463 | 46214 |
| 64 | 12596 | 27166 | 39832 | 45744 |
| 128 | - | 22386 | 37955 | 44847 |
| 256 | - | 15196 | 34400 | 42838 |

Table 6: GLA Model SP Throughput/GPU (tokens/sec) on 1B-32k

| GPU Number | Method | | | |
|---|---|---|---|---|
| | LASP1 | LASP2 | ZeCO | GLA (baseline) |
| 8 | 27014 | 42946 | 47369 | 49633 |
| 16 | 23302 | 41190 | 46209 | 49058 |
| 32 | 18129 | 39669 | 45091 | 47980 |
| 64 | 12268 | 37485 | 44468 | 48230 |
| 128 | - | 33327 | 43278 | 47848 |
| 256 | - | 25402 | 40967 | 46588 |

