# OpenReview forum: "ZeCO: Zero-Communication Overhead Sequence Parallelism for Linear Attention"
_NeurIPS.cc/2025/Conference — NeurIPS 2025 poster_

### Official Review · Reviewer_LbPt · 2025-06-30

**Clarity:** 3
**Significance:** 3
**Originality:** 3
**Rating:** 5
**Confidence:** 3

**Summary:**

In this paper, the authors propose a new sequence parallel method for linear attention. While there are a lot of sequence parallel methods for full attention, linear attention has less computation and therefore communication will be a bottleneck. The proposed ZeCO algorithm specifically optimizes for communications. The method is proved both in theory as well as in practice.

**Questions:**

1. What is the DP performance (i.e. fsdp, zero) in figure 1?
2. What is a full attention performance (i.e.  Ulysse) in figure 1?
3. Is the proposal method easily applied to VLM (vision language model)?

**Ethical Concerns:**

["NO or VERY MINOR ethics concerns only"]

**Final Justification:**

The rebuttal addressed my concerns.

**Limitations:**

yes

**Quality:**

3

**Strengths And Weaknesses:**

Strengths:
1. The paper is well written.
2. The comparison w/ LASP and LASP2 shows advantage of the proposal method

Weaknesses:
1. Could you provide a loss comparison before and after applying ZeCO to show that there is no numerical issue when applying the proposal method?
2. Could you provide a profile to show how the overlapping of intra-device computation and inter-device communication look like in practive? Sometimes, the overhead of the overlapping itself may be larger than the benefit of overlapping, so would like to double check to make sure the communication and computation takes similar time and therefore overlapping makes sense.

---

> ### Author Rebuttal · Authors · 2025-07-31
>
> > Weakness1: When applying the ZeCO Method, numerical issues need to be checked.
>
> **A:** The reviewers' requests have made our work more rigorous. To check the numerical issues, we conducted thorough checks ranging from the numerical values of decay and multiplication in the *All-Scan*, the numerical values of *Operator*, to the overall training loss of the *Model*. The results indicate that our method demonstrates good numerical stability.
> - **All-Scan:** We compared the results of All-Scan with those of All-Gather + Prefix Sum in the LASP-2 method. The numerical difference between the two is on the order of 1e-6.
> - **Operator:** We compared the output and gradient numerical values between the ZeCO method and the full-sequence single-GPU computation, using the ratio-error calculation method from flash-linear-attention. For the operators, the output and gradients of q, k, and v have zero error, while the gradient error of gk is on the order of 1e-6.
> - **Model:** We ran a 1B model using three methods: Ulysses sequence parallelism, data parallelism, and ZeCO sequence parallelism. The model was trained on 50B tokens in SlimPajama. The losses were 2.38, 2.32, and 2.31, respectively. Therefore, the overall training of the model is numerically stable.
>
> > Weakness2: Double Check for Overlapping.
>
> **A:** We are honored to receive such professional advice. In fact, we have also considered this point. Following your suggestion, we have obtained the profiler trace. However, due to rebuttal guidelines, we provide a textual description instead.
>
> - **Profiler Analysis:**
> We provide a timing breakdown comparing the overlapped and non-overlapped execution modes of ZeCO. The table below shows the communication and computation durations extracted from the torch profiler:
>
> | Mode        | Comm Start (ms) | Comm End (ms) | Comp End (ms) |
> |-------------|------------------|----------------|----------------|
> | Non-Overlap | 2769.915         | 2770.020       | 2770.476       |
> | Overlap     | 2542.205         | 2542.384       | 2542.675       |
>
> | Mode        | Comm Duration (ms) | Comp Duration (ms) | Total Duration (ms) |
> |-------------|----------------|----------------|-------------|
> | Non-Overlap | 0.105          | 0.456          | 0.561       |
> | Overlap     | 0.179          | 0.470          | 0.470       |
>
> As seen above, the total step latency is reduced from 0.561 ms to 0.470 ms. This confirms that communication and computation are indeed overlapped effectively, leading to a reduction in latency.
>
> - **Theoretical Analysis:** In practice, the number of thread blocks launched by All-Scan is very small (around 8 to 16), which is far fewer than the number of SMs available on the GPU. Based on this, we aim to fully utilize the SMs and therefore consider it feasible to partition streams to overlap communication and computation. To ensure that communication completes as quickly as possible and to minimize contention and waiting, we assign a higher priority to the Communication Stream.
>
> > Question 1 and 2: DP performance and full attention performance.
>
> **A:** Thank you for your valuable questions. To address your concerns, we have added the experimental results for both the GLA-DP baseline and the Full Attention (Ulysses) model in the table below:
>
> | Methods\ GPU number | 8 (256K) | 16 (512K) | 32 (1M) | 64 (2M) | 128 (4M) | 256 (8M) |
> |---------------------|----------|-----------|---------|---------|----------|----------|
> | LASP1               | 27014    | 23302     | 18129   | 12268   |          |          |
> | LASP2               | 42946    | 41190     | 39669   | 37485   | 33327    | 25402    |
> | **ZeCO (ours)**               | **47369**    | **46209**     | **45091**   | **44468** | **43278**  | **40967**  |
> | GLA-DP       | 49633    | 49058     | 47980   | 48230   | 47848    | 46588    |
> | Full Attn (Ulysses) | 3724     | 1885      | 952     |         |          |          |
>
> It should be noted that LASP1 is not scalable beyond 64 GPUs in our current setup due to its high communication overhead — it becomes prohibitively slow to complete under practical time constraints.
> Regarding Full Attention (Ulysses), since 1B model has a hidden width of 2048, the head number is set to 16 or 32, the parallelism of Ulysses is limited by the number of heads (The maximum value of SP is 32).
>
> We hope this addition provides a more complete comparison across baselines and attention mechanisms.
>
> > Question 3: Is the proposal method easily applied to VLM (vision language model)?
>
> **A:** Yes, we are pleased to inform you that our method can be easily applied to hybrid models in vision-language models (VLMs). Currently, there is considerable research in the industry on deploying linear hybrid models for multimodal training. For example, Nemotron-H and Minimax-01 [1][2] both have versions of vision-language models. Their scales range from 8B to over 400 billion parameters, demonstrating that these are becoming mature solutions in the industry. Among them, Nemotron uses Mamba-2 for its linear layers, while Minimax-01 uses Lightning Attention. Our ZeCO method is compatible with both types of attention.
>
> [1] Blakeman, Aaron, et al. "Nemotron-h: A family of accurate and efficient hybrid mamba-transformer models." arXiv preprint arXiv:2504.03624 (2025).
>
> [2] Li, Aonian, et al. "Minimax-01: Scaling foundation models with lightning attention." arXiv preprint arXiv:2501.08313 (2025).

---

### Official Review · Reviewer_Nor9 · 2025-07-01

**Clarity:** 3
**Significance:** 2
**Originality:** 2
**Rating:** 4
**Confidence:** 3

**Summary:**

The paper introduces a novel method called ZeCO to address the communication bottlenecks in sequence parallelism (SP) for linear attention models, particularly relevant for training large language models (LLMs) on ultra-long sequences (e.g., 1M tokens). Contributions: 1. Enables minimal communication. 2. achieves zero communication overhead and minimal extra computation/I/O.

**Questions:**

1. My concern is that linear attention typically will lead to obvious accuracy drop. So can authors explain the impacts on model accuracy?
2. Can authors explain how well the performance improvements when this method is generalized to other linear attention variants?

**Ethical Concerns:**

["NO or VERY MINOR ethics concerns only"]

**Limitations:**

yes

**Paper Formatting Concerns:**

No format issue.

**Quality:**

3

**Strengths And Weaknesses:**

Strengths:
1.  The paper rigorously proves the communication optimality of the proposed ZeCO approach, showing it achieves the theoretical lower bound in communication volume and latency.
2. Experiments are run at scale and compare ZeCO against strong baseline.

Weaknesses:
1. it's unclear how well the performance improvements generalize to other linear attention variants.
2. The method doesn’t generalize to full attention or hybrid approaches, Besides the accuracy of linear attention is typically bad.

---

> ### Author Rebuttal · Authors · 2025-07-31
>
> > Weakness 1: performance improvements generalize to other linear attention variants.
>
> **A:** When scaling up to 7B and 13B, we did not use the GLA model; instead, we **employed a method similar to HGRN2**, where the data-dependent decay generation is tied with the key vectors. This can be regarded as a “linear model variant.” In terms of performance, ZeCO still guarantees linear scalability, with throughput per GPU nearly approaching that of data parallelism (DP). Therefore, it is evident that our method is applicable to variants of linear attention.
>
> **7B model**
> | method / GPUs (seq len)      | 16 (64K) | 32 (128K) | 64 (256K) | 128 (512K) |
> |------------------|----------|-----------|-----------|-------------|
> | dp               | 3770     | 3685      | 3659      | 3629        |
> | **ZeCO (ours)**             | **3682**     | **3588**      | **3533**      | **3459**        |
> | lasp2            | 3083     | 2595      | 2276      | 1975        |
>
>
> **13B model**
> | method / GPUs (seq len)       | 16 (64K) | 32 (128K) | 64 (256K) | 128 (512K) |
> |------------------|-----------|------------|------------|--------------|
> | dp               | 2277      | 2221       | 2181       | 2181         |
> | **ZeCO (ours)**             | **2242**      | **2168**       | **2130**       | **2095**         |
> | lasp2            | 1622      | 1584       | 1502       | 1389         |
>
> > Weakness 2: The method doesn’t generalize to full attention or hybrid approaches. The accuracy of linear attention is typically bad.
>
> **A:** We also conducted **experiments on 7B and 13B hybrid models**. We want to emphasize that our method applies to the linear attention components within hybrid models, where linear models constitute a dominant portion.
>
> Next are the experimental results for the hybrid model (where we use the Ulysses method for full attention). Although full attention becomes a computational bottleneck at longer sequence lengths, **ZeCO still demonstrates the best performance among sequence parallelism methods.**
>
> All experiments use TP=8, with SP size = (num GPUs / 8). Sequence length fixed at 32K. Activation checkpointing enabled.
>
> **7B Model**
>
> |GPUs(Seq len) / Method        | Hybrid-dp | **Hybrid-ZeCO (ours)** | Hybrid-lasp2 | Full |
> |----------------|-----------|-------------|---------------|------|
> |32 (128K) | 3552      | **2941**        | 2402          | 1298 |
>
> **13B Model**
>
> |GPUs(Seq len) / Method        | Hybrid-dp | **Hybrid-ZeCO (ours)** | Hybrid-lasp2 | Full |
> |----------------|-----------|-------------|---------------|------|
> |40 (160K)| 2135      | **1692**        | 1496          | 697  |
>
> > Question 1: Impacts on model's accuracy.
>
> **A:** First, we propose a sequence parallelism method that enables training on ultra-long sequences, but **this method is independent of the model’s performance quality**. However, this is a very good question, because pure linear models themselves have limited performance. Therefore, the method needs to serve better-performing models in order to be valuable. We summarize the value of our ZeCO SP as follows:
> - ZeCO can serve hybrid models, which are one of the current trends in large-scale industrial models. In hybrid models, linear layers—where ZeCO demonstrates its value—constitute a dominant proportion.
> - ZeCO is applicable to the vast majority of cutting-edge linear model variants, such as HGRN2, Mamba-2, RWKV-6, and others. Therefore, ZeCO supports a wide range of linear layers.
> These two points demonstrate that even though pure linear models have performance limitations, the ZeCO method still holds unique value.
>
> > Question 2: Can authors explain how well the performance improvements when this method is generalized to other linear attention variants?
>
> **A:** We understand performance from two aspects: task performance and throughput.
> - **Task performance** as mentioned in Question 1, this method is independent of the model’s performance quality. Therefore, the performance depends on which model ZeCO is applied to.
> - **Throughput** At the current stage, our method is applicable to data-independent decay models such as RetNet, RWKV-5, and Lightning Attention, as well as data-dependent decay models like GLA, Mamba-2, RWKV-6, and HGRN2. Since the underlying operators of these models are all compatible with the GLA operator, there is no difference in throughput among them.

---

> > ### Comment · Reviewer_Nor9 · 2025-08-01
> > **response to rebuttal**
> >
> > I confirm the response. I am satisfied with the response.

---

> ### Author Response · Authors · 2025-08-04
> **Response to Reviewer Nor9**
>
> Dear Reviewer Nor9,
>
> We greatly appreciate your positive feedback on our response. Your constructive suggestions provide excellent guidance for presenting this work to the broader community. Thank you for taking the time to share these valuable insights with us.

---

### Official Review · Reviewer_NCSM · 2025-07-03

**Clarity:** 3
**Significance:** 3
**Originality:** 3
**Rating:** 5
**Confidence:** 1

**Summary:**

The paper introduces ZeCO, a sequence parallelism (SP) method that sharply reduces communication bottlenecks in training linear attention models on long input sequences. ZeCO is built around a new communication primitive called All-Scan, which uses pipelined communication to reduce latency when sending state information from a predecessor to a successor devices. They show—empirically and theoretically—that this achieves a near-optimal communication volume that hides communication latency by overlapping with local computation. They demonstrate strong empirical runtime performance on 1B-scale linear attention models in compared to other methods.

**Questions:**

Are there other applications for All-Scan beyond this paper?

**Ethical Concerns:**

["NO or VERY MINOR ethics concerns only"]

**Final Justification:**

The author's responses were helpful and clarifying, and did not impact my final score.

**Limitations:**

I would recommend that the authors add an explicit "Limitations" section where they can summarize their assumptions and hardware constraints.

**Quality:**

4

**Strengths And Weaknesses:**

# Strengths

A key concern about linear attention models is their amenability to modern hardware. This work represents a novel systems-level contribution resolves a key communication bottleneck, making it easier to scale such models to longer contexts.

The paper is well-structured, and ideas are presented in a clear and concise manner. Figure 2 is a great visualization of the approach and its advantages.

The empirical results are strong and demonstrate compelling scaling behavior, benchmarked against alternative approaches.

# Weaknesses

The approach is limited in scope to linear attention and focuses on GLA models in particular. Can something similar to ZeCO be adapted to quadratic attention models? Are there common subquadratic transformers that cannot be implemented using ZeCO?

Code is not included in the submission. What is the timeline for open-sourcing the code?

In the optimality analysis, I would recommend including stating your assumptions (e.g. which costs are assumed to be negligible) more directly at the start of the analysis.

Minor points:
- L15: “optimaity” —> “optimality”
- Algorithm 1: “comunication” —> “communication”

---

> ### Author Rebuttal · Authors · 2025-07-31
>
> > Weakness 1: Limited to GLA Model. Can ZeCO-Like Method be adapted to quadratic attention? Any subquadratic attention cannot adapt to ZeCO?
>
> **A:** The reviewer’s question is very insightful. In summary, **first**, the ZeCO method applies to linear models with materialized chunk-wise algorithm and Prefix–Scannable Model [1][2], such as GLA, Mamba-2, HGRN2, and Lightning Attention. Furthermore, it can be used in hybrid models that include these linear attention layers. **Second**, the ZeCO method is designed specifically to fully unleash the performance of linear attention across multiple GPUs. Therefore, unfortunately, we are unable to apply ZeCO-like methods to quadratic attention.
>
> **Which models can use ZeCO?**
> - First, the algorithm needs to follow the chunk-wise parallelism paradigm. For example, the materialized state version of the GLA operator. The algorithm first computes the state for each chunk, which then enables a communication pattern to globally synchronize the initial states across devices.
> - Second, the algorithm must possess the property of being a prefix-scannable model. This allows all devices to perform parallel computations on the states that need to be updated for their local sequences before communication.
>
> **Negative Examples** In the Mamba algorithm, due to the materialization of each chunk’s state consuming excessive GPU memory, only parallel scan is possible, which does not satisfy chunk-wise parallelism. Therefore, ZeCO cannot be applied. For vanilla RNNs, ZeCO cannot be used because they do not possess the prefix-scannable model property.
>
> **Positive Examples** Data-independent attention models such as RetNet and Lightning Attention, as well as data-dependent attention models such as HGRN2, Mamba-2, and GLA, can all be implemented using the GLA operator. Therefore, ZeCO can be applied to all these models.
>
> **Future Works** Models that perform state updates based on matrix transformations, such as DeltaNet and DeltaProduct, satisfy both the materialized chunk-wise algorithm and the prefix-scannable model properties, making them theoretically compatible. However, we still need to implement the matrix transformation All-Scan operation, so adaptation will be carried out in the future.
>
> > Weakness 2: What is the timeline for open-sourcing the code?
>
> **A:** Our communication algorithm is based on nvshmem. We found that Triton-distributed [3][4] enables easier implementation and development. However, this repository is currently undergoing large-scale modifications and adaptations. This month, Triton-distributed has become more stable. We plan to adapt our implementation to the stable version and open-source it for the community’s reference and use. Furthermore, in the future, we will consider integrating this operator into the Triton-distributed project as a fundamental collective communication primitive, contributing to the infrastructure community.
>
> > Weakness 3: I would recommend including stating your assumptions more directly at the start of the analysis.
>
> **A:** This is a very helpful suggestion. We have noticed that another reviewer also raised some questions regarding the analysis of optimality. The reason we included such an analysis in the paper was to provide a quantitative analysis and comparison approach to algorithm design for sequence parallelism in linear models. We acknowledge that the lack of clearly stated assumptions has caused some confusion during reading, and we sincerely apologize for this.
>
> In future versions, we will make the following revisions:
> - Definitions: We will provide a clear definition of linear model algorithms. The algorithms we adapt and analyze should satisfy (1) materialized chunk-wise algorithm and (2) prefix-scannable model properties.
> - Assumptions: We will state the basic assumptions used in analyzing communication and computational complexity, such as 1)Ignoring lightweight computations within the communication kernel.  2)Computations that can be neglected compared to I/O and operator computation overhead.
>
> > Writing Suggestion
>
> **A:** Thank you for your writing suggestions. We will modify them in the revised version.
>
> > Question 1: Are there other applications for All-Scan beyond this paper?
>
> **A:** In this paper, besides the ZeCO sequence parallelism method, the introduction of the All-Scan collective communication primitive is also one of our contributions. Currently, this novel collective communication is only applied to support the ZeCO sequence parallelism method. However, we will assist the community in improving the All-Scan communication primitive to make it a reusable “building block” for the community. We look forward to this communication primitive playing a greater role in the future.
>
> [1] Yang, Songlin, et al. "Gated linear attention transformers with hardware-efficient training." arXiv preprint arXiv:2312.06635(2023).
>
> [2]Yau, Morris, et al. "Sequential-Parallel Duality in Prefix Scannable Models." arXiv preprint arXiv:2506.10918 (2025).
>
> [3] Zheng, Size, et al. "Triton-distributed: Programming Overlapping Kernels on Distributed AI Systems with the Triton Compiler." arXiv preprint arXiv:2504.19442 (2025).
>
> [4] Zheng, Size, et al. "Tilelink: Generating efficient compute-communication overlapping kernels using tile-centric primitives." arXiv preprint arXiv:2503.20313 (2025).

---

> > ### Comment · Reviewer_NCSM · 2025-08-05
> >
> > Thank you for your detailed response. I will maintain my score.

---

> > > ### Author Response · Authors · 2025-08-07
> > > **We are very glad to address your concerns in our response.**
> > >
> > > Dear Reviewer NCSM,
> > >
> > > We are very glad to address your concerns in our response. We hope these answers help you gain a better understanding of our work. From your review comments, we can feel your support for our research. Our work spans multiple areas including collective communication, attention kernels, and parallelism, so we have made an effort to introduce the background of each aspect in accessible language throughout the paper. We welcome any further questions you may have and hope that you feel more confident when evaluating our work.

---

### Official Review · Reviewer_8i2q · 2025-07-09

**Clarity:** 2
**Significance:** 2
**Originality:** 3
**Rating:** 3
**Confidence:** 3

**Summary:**

This paper introduces ZeCO (Zero Communication Overhead), a sequence parallelism (SP) method designed to overcome communication bottlenecks in training large language models (LLMs) with linear attention mechanisms for very long sequences. Existing SP methods for linear attention suffer from substantial communication overhead, hindering efficient scaling across multiple devices. To address thiz challenge, the authors propose "All-Scan," a new collective communication primitive that provides each SP rank with the necessary initial operator state while maintaining a minimal communication footprint, effectively eliminating communication overhead. By leveraging All-Scan, ZeCO provides nearly optimal performance by introducing negligible time and space overhead. By using experimental evaluation, the authors show that ZeCO achieves a 60% speedup over state-of-the-art SP methods on 256 GPUs for an 8M sequence length, and exhibiting near-linear scalability for long sequence training.

**Questions:**

1) How many GPUs can ZeCO scale to while maintaining near-linear throughput scaling for a 1B model?
2) Can the global state be divided along both $d_k$ and $d_v$ dimensions ?
3) How does ZeCO scale with larger models? While the authors state that "additional number of parameters does not affect how cost is calculated", the dimensions will differ for larger models. It would be beneficial to see comparable results for larger models to confirm expected scaling behavior.
4) Why is a comparison with full attention SP methods meaningful, given their inherently different workloads? (Referring to Figure 3)
5) Can ZeCO be extended to training hybrid attention models, such as Jamba?

**Ethical Concerns:**

["NO or VERY MINOR ethics concerns only"]

**Limitations:**

yes

**Quality:**

2

**Strengths And Weaknesses:**

Strengths:
1) ZeCO algorithm is intuitive, leverages the characteristic of computation to effectively mask communication time and, importantly, shows superior scaling to long sequences as compared to existing LASP* methods up to 256 GPUs.

Weaknesses:
1) The paper focusses on linear attention models which have not been very successful for most NLP tasks. This limits the impact of this work.

2) I am confused about the "optimality" claim:
2.1)  In order for ZeCO to be "optimal" one would have to prove that any algorithm taking less time than ZeCO would not be able to achieve the required computation (lower-bound proofs). The presented proof doesn't prove this. You might want to reconsider the use of term "optimal" here.
2.2) Even if optimality is used more loosely, I am still not sure about the presented arguments. More specifically
- Line 191. T_{ideal-SP) and t_{overlaped_comp} are key parameters of the algorithm. How does ZeCO make sure that it provides the best choice for overlaped_comp? This is not clear from the writing.
- Line 203-205. when K becomes sufficiently large, the overhead approaches zero. I am not sure about this; since K is upper bounded by $d_k$, in the best case T_{ALL_SCAN} is T_{ALL_SCAN} = \tau(d_k \times d_v) + (p-1) \tau(d_v). For large values of $p$ (which is the focus of this paper is all about), the second term is not negligible; in fact, it can become larger than the first term.
- Line 212. Saying T_extra_comp&IO is negligible is fine, but  this does not imply that ZeCO has made the best possible choices.
- One last point: there is an additional (redundant computation) in ALL_SCAN which I did not find being discussed in the paper.

3) Experiments are limited to single sized model. what happens to in larger models (7B, 13B). Do we see similar scaling.

Writing suggestions:
- Please use standard mathematical notation to improve readability, i.e.., lowercase italic letters for scalars, lowercase bold letters for vectors, and uppercase bold letters for matrices.
- Please make another editing pass. A few examples:
* Line 15: optimality spelling error
* Line 40: this is not paradoxical
* Line 125: "G"lobal
* Line 134: "S"tate
* Line 236: "R"untime

---

> ### Author Rebuttal · Authors · 2025-07-31
>
> > Weakness 1: Limitaion of Pure Linear Model.
>
> **A:**: The reviewer accurately pointed out the current performance limitations of pure linear models. However, we would like to argue that the limitations of pure linear models will not affect our work.
>
> **Hybrid Models and Long Context Problem is Important.** The industry is currently adopting hybrid models, where **linear layers play a dominant role**. Therefore, our method has broad application prospects. Furthermore, the development of fields such as multimodal AI and the recent use of linear models as tokenizers increases the demand for efficient **long-sequence computation**, naturally drawing more attention to sequence parallelism methods for linear models.
>
> > Weakness 2: Reconsider the Optimality Analysis.
>
> **A:** We apologize for the confusion caused by our use of the term "Optimality" here. Our original intention was not to present a mathematically rigorous lower bound, but rather to provide a quantitative analysis and design perspective from an engineering standpoint. The analysis method has practical engineering significance, and we will reconsider how to present this part more clearly.
>
> > Weakness 2.1:Definition of overlapable computation.
>
> **A:** We should provide clearer definitions. In the implementation of Gated Linear Attention (GLA), the authors divide the computation into three operators with distinct functions. The first operator is responsible for materializing inter-chunk states. The second operator computes the intra-chunk attention map. The third operator combines the inter-chunk and intra-chunk results to produce the output. We take the GLA operator as our baseline. Our design and the selection of overlaps are based on these operators as fundamental building blocks. Therefore, the overlap naturally corresponds to the computation of the intra-chunk attention map, as this part does not depend on the state.
>
> > Weakness 2.2: Line 203-205, the term related with split number K and device number p.
>
> **A:** This is a very valuable question. The upper bound of \( K \) is \( d_k \), which is the simplified scenario we presented. However, in practical applications, we also need to consider the number of heads \( H \). The states of all \( H \) heads need to be transmitted. Typically, \( d_k \) and \( d_v \) are 128, but \( H \times d_k \) equals \( D \), where \( D \) is the model dimension. Generally, for models larger than 1B parameters, \( D \) is greater than 2048. Therefore, compared to the number of devices \( p \) used for sequence parallelism, this is a very large number.
>
> > Weakness 2.3: Extra computation and I/O may not the best choice.
>
> **A:** Our design is based on the perspective of global state updates.
> First, to ensure that the All-Scan communication contains a complete decay for the sequence within a device, there must be storage for the \(\gamma\) vector.
> Second, after the global communication, to guarantee that every materialized state undergoes a global update, it is necessary to load the state after All-Scan communication and perform computations with the inter-chunk states; this step cannot be omitted. We will reconsider the way the analysis is presented here.
>
> > Weakness 2.4: additional computation in All-Scan is not discussed
>
> **A:** Thank you for pointing this out. We would like to clarify that the lightweight computation involved in All-Scan communication is included in the time of All-Scan (equation 12) and is inherently negligible to the communication.
>
> This treatment aligns with standard communication analysis of collective communication like ring all-reduce and other bandwidth analysis [1][2].
>
> We will revise the final version to clarify that this lightweight accumulation step is a **standard practice in communication**, and like in prior work, its cost is safely ignored in both theoretical and empirical cost models.
>
> > Weakness 3: Scaling Performance on Larger Models
>
> **A:** Conducting experiments on larger-scale models is a constructive suggestion. To alleviate the memory pressure caused by model scaling, we adapted a simple Tensor Parallelism (TP) to our Sequence Parallelism (SP) code framework. We completed comparisons among LASP-2, ZeCO, and DP, scaling up to a maximum of 128 GPUs. In the experiments, we used a sequence length of 32k per SP/DP rank, with TP=8 and activation checkpointing enabled.
>
> **7B model**
> | method / GPUs (total seq len)      | 16 (64K) | 32 (128K) | 64 (256K) | 128 (512K) |
> |------------------|----------|-----------|-----------|-------------|
> | dp               | 3770     | 3685      | 3659      | 3629        |
> | **ZeCO (ours)**             | **3682**     | **3588**      | **3533**      | **3459**        |
> | lasp2            | 3083     | 2595      | 2276      | 1975        |
>
>
> **13B model**
> | method / GPUs (seq len)       | 16 (64K) | 32 (128K) | 64 (256K) | 128 (512K) |
> |------------------|-----------|------------|------------|--------------|
> | dp               | 2277      | 2221       | 2181       | 2181         |
> | **ZeCO (ours)**             | **2242**      | **2168**       | **2130**       | **2095**         |
> | lasp2            | 1622      | 1584       | 1502       | 1389         |
>
> > Question 1: How many GPUs can ZeCO scale to while maintaining near-linear throughput scaling for a 1B model?
>
> Frankly speaking, “near-linear” is an empirical trend and it is difficult to assign a precise numerical value. However, based on our experimental results, at 256 GPUs, the TGS (tokens per GPU per second) metric already shows some degradation (keeps about 87% of the throughput of data parallelism), but it is still significantly better than LASP-2.
>
> When we mention the concept of “near-linear,” we aim to draw attention within the linear model community. Currently, the definition of linear complexity mostly focuses on single-GPU scenarios, where time cost grows linearly with sequence length. However, in the context of sequence parallelism, the focus should be on **multi-GPU linear complexity**, meaning that if the number of GPUs scales linearly with sequence length, the sequence processing time should remain approximately constant. Under this definition, **it is difficult to refer to  linear models with current sequence parallelism methods as truly "linear".**
>
> > Question 2: Can the global state be divided along both dk and dv dimensions
>
> **A:** This is a very astute observation. **In principle, it is feasible. However, in practice, we usually do not do this.**
> - **In principle**, the decay and add operations for all elements are independent. Therefore, we can naturally partition along both the dk and dv dimensions.
> - **In practice**, there are two reasons why we prefer not to do this. First, computationally, each decay factor \gamma is responsible for the decay of dv number of scalar values. We aim to fully **leverage the GPU’s parallelism** by processing these dv scalars in batches. Second, in terms of communication, we want each transmitted split chunk to be large enough to ensure **communication efficiency**, which is typically on the order of kilobytes (KB).
>
> > Question 3: Scaling Performance
>
> **A**: As you can see, we demonstrated the performance of the ZeCO method on larger models in Weakness 3.
>
> > Question 4: Why compare full attention SP?
>
> **A**: We present the computation and communication costs of full attention in Figure 3. There are **two purposes for this comparison**. First, from an efficiency perspective, it **demonstrates that our model is the most efficient among all** current sequence parallelism methods. Second, through a brief analysis and comparison, we aim to **help potentially confused readers understand the fundamental workload differences** between full attention and linear attention. Therefore, we do not see the need to further compare with full attention in subsequent experiments.
>
> > Question 5: Can ZeCO be extended to training hybrid attention models, such as Jamba?
>
> **A:** Regarding extension to Hybrid Models, the answer is **Yes**, whereas for Jamba, the answer is no. There is a point that needs to be clarified. Our method applies to all chunk-wise parallelism paradigm models [3], such as GLA, HGRN2, Mamba-2, Lightning Attention, RetNet, and so on. Therefore, **for the vast majority of mainstream hybrid models, including Minimax-01, Nemotron-H and Falcon-H1 [4][5][6], our method is compatible.** However, Jamba uses Mamba-1, which lacks a chunk-wise algorithm — this is an inherent limitation that also affects the applicability of sequence parallelism methods like LASP-2 and ZeCO. Referring to Minimax-01, we only need to apply our method to the linear attention layers, which does not affect the use of other parallelism methods in the full attention layers.
>
> **To concretely address this concern, we conducted preliminary experiments on hybrid models and report the results. Please refer our answer to the weakness 2 from reviewer Nor9**
>
> > Writing Suggestion
>
> **A:** Thank you for your writing suggestions. We will modify them in the revised version.
>
> [1] Andrew Gibiansky. Bringing HPC techniques to deep learning. [Online].
>
> [2] Patarasuk, Pitch, and Xin Yuan. "Bandwidth optimal all-reduce algorithms for clusters of workstations." Journal of Parallel and Distributed Computing 69.2 (2009): 117-124.
>
> [3] Yang, Songlin, et al. "Gated linear attention transformers with hardware-efficient training." arXiv preprint arXiv:2312.06635(2023).
>
> [4] Blakeman, Aaron, et al. "Nemotron-h: A family of accurate and efficient hybrid mamba-transformer models." arXiv preprint arXiv:2504.03624 (2025).
>
> [5] Li, Aonian, et al. "Minimax-01: Scaling foundation models with lightning attention." arXiv preprint arXiv:2501.08313 (2025).
>
> [6] Zuo, Jingwei, et al. "Falcon-H1: A Family of Hybrid-Head Language Models Redefining Efficiency and Performance." arXiv, 2025, arxiv.org/abs/2507.22448.

---

> ### Author Response · Authors · 2025-08-07
>
> Dear Reviewer 8i2q
>
> Your review comments are very comprehensive. At a high level, you are concerned with the effectiveness of scaling up and the applicability within hybrid models. On the technical side, you focus on the possible limitations of the algorithm itself as well as the theoretical constraints and shortcomings. We sincerely appreciate the effort you have invested in reviewing our work, which has greatly helped us improve it. In our response, we have provided detailed discussions on the model scalability, hybrid architecture, technical details of communication, and the limitations of our theoretical analysis. We look forward to your feedback to jointly advance the development of the community.

---

> > ### Comment · Area_Chair_zCoH · 2025-08-08
> > **.**
> >
> > Dear reviewer,
> >
> > The discussion period is ending in 24 hours. Please respond to the author rebuttal and engage in discussion. Failure to do so may result in possible penalties.
> >
> > Thanks!
> > Your AC

---

### Author Response · Authors · 2025-08-09
**The Summary of Reviews & Rebuttal**

# The Summary of Reviews & Rebuttal

We are very grateful to all the reviewers for their valuable suggestions during this review process. Here, we summarize the reviewers’ evaluations of our work, the issues addressed in this round of responses, and the extended explorations we have carried out on the work.

---

## Summary of this work

In this paper, we introduce ZeCO (Zero Communication Overhead) sequence parallelism for linear attention models, a new SP method designed to overcome these limitations and achieve end-to-end near-linear scalability for long sequence training.

Our contributions are as follows:

1. We propose ZeCO, a sequence parallelism method that provides a **systematic design** for linear attention sequence parallelism. (1)We introduce a **new collective communication operation**, All-Scan. (2)We **schedule overlapping** of communication and computation overheads to maximize resource efficiency. (3)We reduce the additional I/O and computation overhead to **less than 1%** of the original cost.

2. We theoretically analyze the overhead of our method and compare it with existing approaches. Our method achieves the **best theoretical performance** among current solutions.

3. Empirically, we conduct experiments demonstrating that the **All-Scan** collective communication achieves a **3.9× speedup** over existing communication methods. The sequence parallel operator reduces additional overhead **by a factor of nine** compared to previous methods. Our model scales sequence parallelism up to **256 GPUs** with sequence lengths of **8 Million**. From the performance trends, our method is the **first and currently the only sequence parallelism approach achieving nearly linear scaling.**

---

## Review comments

**About the ZeCO Algorithm:** Reviewer '8i2q' considers the algorithm to be **intuitive** and it **leverages the characteristic of computation**. Reviewer 'NCSM' highlights the algorithm’s **amenability to modern hardware** and regards it as a **novel systems-level contribution**.

**About Experiments:** Reviewer '8i2q' 'NCSM' 'Nor9' mentioned that our experimental results **demonstrate excellent scalability.** Reviewer 'NCSM' 'Nor9' 'LbPt' mentioned that our method shows **advantages compared to other approaches.**

**About Paper Writing:** Reviewer 'NCSM' mentioned that our paper is **well-structured** and the ideas are presented **in a clear and concise manner.** 'Nor9' said that the paper is **well written.**

---

## Concerns we addressed

**Our method can be applied to hybrid models.** Reviewers expressed concerns about whether our method is limited to pure linear models. Through theoretical analysis and empirical validation, we demonstrate that our method is **applicable to both linear models and hybrid models.** This significantly broadens the applicability and impact of our approach.

**Our method can be applied to a broader range of linear model variants.** Reviewers raised concerns about whether our method can only be applied to sequence parallelism for Gated Linear Attention. We clarify the underlying principles and the scope of applicability of our method. For example, it can be **adapted to various linear attentions** such as RetNet, RWKV5, HGRN2 and Mamba-2.

**Our method can scale to larger models.** The reviewers noted that the experiments presented in the paper were conducted on 1B-scale models and inquired whether the same scalability holds for 7B or 13B models. Through empirical evaluation, we demonstrate that our method **remains effective on 7B and 13B models**, even showing **greater advantages.**

**Our method has been rigorously validated numerically.** Reviewers expressed concerns about whether our communication and computation kernels align with the original operator’s precision. We addressed this concern through thorough **validation of communication, operator, and model accuracy.** Our operator has undergone rigorous precision verification.

**Other technical details.** Reviewers also raised concerns regarding the communication partitioning scheme and the effectiveness of overlapping. We addressed these issues through both theoretical analysis and empirical evaluation. Our partitioning strategy carefully considers hardware characteristics and scalability, and the effectiveness of the overlapping method has been rigorously validated through experiments.

---

### Note · Authors · 2025-08-16

Dear Area Chair,

---
# Key Achievement

**Key Achievement:** In the discussion stage, three of four reviewers replied to our rebuttal. Each offered a **concise and positive response**, stating that our answers were **detailed, satisfactory, and successfully resolved their concerns**.

---
# Summary of Reviewer Response

We respectfully request your careful consideration given this **positive reviewer response** to our additional experimental evidence and further theoretical analysis.

**Model Scaling** Our approach successfully validated on **7B and 13B** models using **128 GPUs**. It outperforms current state-of-the-art (SOTA) methods, delivering **75% and 50% greater throughput** on the 7B and 13B models, respectively.

**Hybrid Model Application** The method was also adapted for hybrid 7B and 13B models. This confirmed its **SOTA effectiveness** within the Hybrid architectures prevalent in industrial settings.

**Linear Attention Variants Application** The generality of our approach is confirmed by its compatibility beyond just Gated Linear Attention (GLA). It seamlessly adapts not only to GLA but also to a **broader family of advanced linear attentions**, including RetNet, Ligntning Attention, RWKV5/6, HGRN2, and Mamba-2.


**Numerical Stability** We have verified that our method possesses **high numerical stability**. The numerical values at each level are comparable to those of the single vanilla computation.

---
# Reviewers' Feedback

**Reviewer 8i2q:** (No response).

**Reviewer NCSM:** Thank us for our **detailed** response. He/She will maintain his/her score (Rating 5).

**Reviewer Nor9:** He/She confirms the response. He/She is **satisfied** with the response.

**Reviewer LbPt:** The rebuttal **addressed** his/her concerns.

---

Our method is the first to enable sequence parallelism for linear models with nearly zero additional overhead. Furthermore, we introduce a novel collective communication primitive. We believe this work holds significant academic and industrial value.

Best regards,

The Authors

---

### Decision · Program_Chairs · 2025-09-17

**Decision:**

Accept (poster)

**Comment:**

This paper proposes an efficient sequence-parallel algorithm for training (certain types of) linear attention models that reduces communication. This involves introducing a new "All scan" communication primitive, which makes it possible to efficiently scale linear attention to long (1M) sequences

On the plus side, the approach is sensible, the paper is well-written, and the empirical experiments support the main claims of the paper. The authors moreover plan to open source their code, which will be helpful to the community.

On the negative side, there were some concerns with regard to: the approach only being relevant for particular types of linear attention models; whether the use of "optimal" is theoretically sound; whether the approach would scale to larger models/GPUs. The authors addressed some of these issues during the rebuttal. I especially encourage the authors to consider the presentation of section 3.3 in light of feedback from reviewer 8i2q.

Overall, I think this is a valuable contribution to the community.